# How to Spin an Object: First, Get the Shape Right

## Abstract

We present unPIC, a method for generating novel 3D-consistent views of an object from a single image. Given one input view, unPIC produces a full spin of the object around its vertical axis, a process that is typically a precursor for reconstructing the object in 3D. Our key idea is to predict the object's underlying 3D geometry from the input image *before* predicting the textured appearance of the novel views. To this end, unPIC consists of two modules: a multiview geometry *prior*, followed by a multiview appearance *decoder*, both implemented as diffusion models but trained separately. During inference, the geometry serves as a blueprint to coordinate the generation of the final novel views, thus enforcing consistency across the object's 360-degree spin. We introduce a novel pointmap-based representation to capture the geometry, with one key advantage: it allows us to obtain a 3D point cloud directly as part of the view-synthesis process, rather than a post-hoc step. Our modular, geometry-driven framework outperforms leading methods like InstantMesh, EscherNet, CAT3D, and Direct3D on novel-view quality, geometric accuracy, and multiview-consistency metrics. Furthermore, unPIC shows strong generalization to challenging, real-world captures from datasets like Google Scanned Objects and the Digital Twin Catalog. We provide code and model weights at: `http://github.com/unpic-iclr-2026/unpic`.

## 1 Introduction

Recovering 3D appearance from a single image is a hard, underspecified problem (Hartley & Zisserman, 2003; Li et al., 2024b). A process that generates multiple views from a monocular image of a subject can serve as a useful precursor for 3D reconstruction (Liu et al., 2023; Gao et al., 2024; Kong et al., 2024). Such a novel-view prior is commonly implemented as a diffusion denoising probabilistic model (Ho et al., 2020) to capture a distribution of valid 3D outputs. A common pitfall, however, is that most implementations conflate the prediction of the unseen geometry and texture, making the output space complex and the task potentially harder. This approach stands in contrast to traditional 3D graphics pipelines, which have long enforced a separation of concerns: a 3D mesh is first constructed (geometric modeling), and only then 'textured' and 'lit' with materials and shaders (appearance generation).

To disentangle the prediction of geometry and texture, we introduce a modular framework for single-image-to-3D generation. We model two distributions that permit sequential sampling, $p(\text{geometry} \mid \text{image})$ and $p(\text{appearance} \mid \text{geometry}, \text{image})$, thus factorizing the target distribution $p(\text{3D} \mid \text{image})$. Since image to 3D is a one-to-many mapping, modeling it as the composition of two probabilistic maps can improve the range and accuracy of the realized outputs. Our hierarchical approach is analogous to the unCLIP approach behind DALL-E 2 (Ramesh et al., 2022), a popular model for text-to-image synthesis: they first map CLIP text embeddings to image embeddings using a probabilistic *prior*, then *decode* the image embeddings to pixels. Our image-to-3D model, called unPIC (undo-a-Picture), can also be seen as downstream of unCLIP. Our pipeline allows separate training of the geometry and appearance modules, and flexibility in sampling multiple geometries or multiple textures for the same geometry (see Figure 1).

A key enabler of this design is a new dense geometric representation: CROCS (Camera-Relative Object Coordinates). CROCS encodes per-pixel 3D coordinates inside a unit cube anchored to the source camera, yielding pointmaps (aka coordinate maps) that are predictable across all 360° target

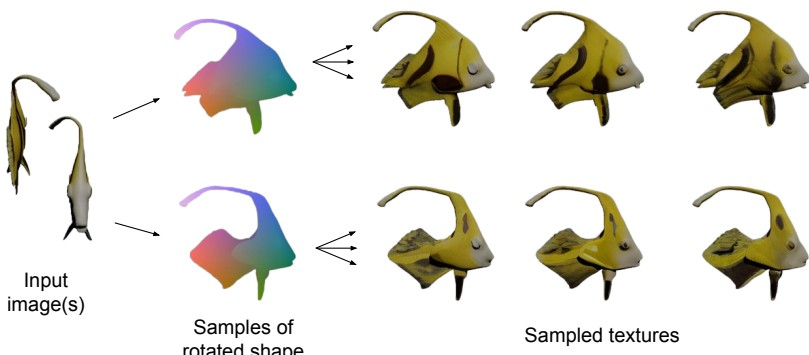

Input image(s)

Samples of rotated shape

Sampled textures

Figure 1: **Our hierarchical probabilistic approach (unPIC).** Generating 3D from a few images (left) is an underspecified, one-to-many task. Our model addresses this with a two-stage probabilistic framework: a prior samples multiple distinct 3D geometries (middle), represented here as pointmaps. For each sampled geometry, a decoder then synthesizes a variety of plausible textures and appearances (right), resulting in a diverse set of novel views. This hierarchical approach allows our model to explore a broader range of shapes and textures consistent with the input image(s).

views. The prior module samples a set of CROCS pointmaps that both stitch into a coherent 3D point cloud and serve as a blueprint for appearance synthesis. Conditioning the appearance decoder on CROCS produces a complete set of novel views, which assemble into a colored point cloud— achieving *direct 3D via view synthesis* without a separate reconstruction stage.

We validate this design in four steps: (i) we show that CROCS is an effective conditioning signal for novel-view synthesis compared to depth, image features, or other pointmaps (Section 4.1); (ii) we demonstrate that CROCS is predictable from a single view by a multiview diffusion prior (Section 4.1); (iii) we evaluate the full unPIC pipeline on novel-view quality, 3D geometric accuracy, and multiview consistency against strong baselines including CAT3D, EscherNet, One-2-3-45, OpenLRM, Free3D, InstantMesh, Direct3D, and MeshLRM (Sections 4.2 and 4.3); and (iv) we ablate the hierarchy by removing geometry from training and inference, observing substantial drops relative to the full model (Section 4.4).

## 2 RELATED WORK

### 2.1 BACKGROUND

**Classic novel view synthesis (NVS):** Scene-specific methods such as NeRF (Mildenhall et al., 2021) and 3D Gaussian Splatting (Kerbl et al., 2023) fit explicit geometry but require multiple consistent input views. When only one image is available, these methods need a learned prior to predict what the scene looks like from alternative views. Such priors include pixelNeRF (Yu et al., 2021) and LRM (Hong et al., 2024). They rely on spatially arranged representations to enable feed-forward NVS, but are deterministic, and suffer from averaged/blurry reconstructions.

**Diffusion-based 3D generation:** Early text-to-3D methods such as DreamFusion (Poole et al., 2023) and Score Jacobian Chaining (Wang et al., 2023) introduced the use of diffusion priors for iterative 3D generation. But they optimize per scene and can suffer from multi-face (Janus) artifacts. Subsequent image-to-3D models like Zero-1-to-3 (Liu et al., 2023) and ZeroNVS (Sargent et al., 2024) train 3D-aware diffusion models from single images, enabling feed-forward inference, typically followed by an optimization step to produce a final 3D representation. This "generate then optimize" paradigm has become a dominant strategy for 3D reconstruction.

**Multiview diffusion for NVS:** Recent methods such as EscherNet (Kong et al., 2024), Free3D (Zheng & Vedaldi, 2024), ReconFusion (Wu et al., 2024a), and CAT3D (Gao et al., 2024) denoise K views simultaneously to exchange information across images and promote consistency. Nevertheless, their lack of geometric grounding can result in generated images that are individually plausible but collectively represent an impossible 3D object. Efforts to mitigate this, such as 3DiM's stochastic conditioning (Watson et al., 2023), still lack direct geometric control.

## 2.2 MOTIVATION

**Geometry as an input:** ControlNet (Zhang et al., 2023) popularized the use of pixel-aligned hints to condition and guide a diffusion model. Other successful applications include motion-brush animation (Niu et al., 2024) and object-pose-conditioned scene/layout generation (Jabri et al., 2024; Wu et al., 2024c). These non-hierarchical approaches are especially tempting given that diffusion models can be trained with conditioning dropout to enable classifier-free guidance (Ho & Salimans, 2022). At inference time, the model can cope with not having any geometric information, and still sample from the unconditional distribution.

**Geometry as an output:** Pretrained diffusion models have been shown to yield latents that contain useful, implicit 3D information (El Banani et al., 2024; Dutt et al., 2024). More explicitly, works like He et al. (2024); Ke et al. (2024); Fu et al. (2024) output dense annotations such as depth and surface normals, but typically annotate monocular images rather than predict novel views. They lack geometric consistency when applied to different views of a scene. We show that our pointmap-based representation (CROCS) facilitates explicit, 3D-consistent geometry prediction.

**Hierarchical generation:** Bringing together the *use* and *prediction* of geometry as an input and output, unPIC first predicts novel-view geometry with a dedicated prior, then uses it to condition a separate appearance decoder. Aside from Motion-I2V (Shi et al., 2024), few works have explored sampling intermediate geometric features and decoding them in lieu of true or user-provided inputs. We show this hierarchical generation is not only possible, but desirable. Sampling from a prior ensures the geometric conditioning is realistic for the given example, diverse (e.g., different 3D shapes corresponding to a 2D image), and obtainable without human effort. In contrast to end-to-end training, using a dedicated appearance decoder trained on ground-truth geometry helps maintain a separation of concerns, which in turn promotes the diversity of predictions at each stage.

## 2.3 CURRENT APPROACHES

**Geometry-supervised image-to-3D.** Methods such as One-2-3-45 (Liu et al., 2024) (SDF via cost volumes) and InstantMesh (Xu et al., 2024b) (triplane to mesh with depth/normal supervision) incorporate explicit geometric signals and return explicit surfaces. Closer to our work, Bolt3D (Szymanowicz et al., 2025) and CRM (Wang et al., 2024b) use pointmaps as their geometric representation; however, geometry and appearance are predicted jointly (Bolt3D) or appearance-first (CRM), rather than unPIC's geometry-first approach. Two recent approaches, TRELLIS (Xiang et al., 2024) and CLAY (Zhang et al., 2024), also use a geometry-first approach, but rely on specialized 3D VAEs rather than our 2D-pretrained VAE (from Stable Diffusion 1.4).

A separate line of work focuses on untextured mesh generation (e.g., CraftsMan3D (Li et al., 2025), Direct3D (Wu et al., 2024b)), emphasizing native 3D generation and shape quality.

**Pointmaps:** DUSt3R (Wang et al., 2024a) predicts dense pointmaps to establish 3D correspondence between images, but needs global alignment and does not model unseen views. VGGT (Wang et al., 2025) proposes viewpoint-invariant pointmaps with per-scene normalization. SpaRP (Xu et al., 2024a) uses a diffusion model to estimate NOCS pointmaps (Wang et al., 2019) for the given views; in the single-image-to-3D setting, SpaRP would predict a single NOCS image to annotate the source view only, since the model does not use pointmaps to hallucinate novel-view geometry.

SweetDreamer (Li et al., 2024a) is closer to unPIC in its use of 'Canonical Coordinate Maps' to predict novel-view geometry, but still relies on the NOCS-assumption that "all objects within the same category adhere to a canonical orientation"; it further relies on optimization (score distillation) to produce the final 3D reconstruction. Our CROCS differs by being camera-relative unlike Sweet-Dreamer, and normalized consistently to $[0, 1]$ unlike VGGT. CROCS yields predictable statistics across target viewpoints, and enables direct point-cloud extraction during view synthesis.

# 3 METHOD

## 3.1 HIERARCHICAL GENERATION

unPIC is a hierarchical generative model that takes a 2D image and predicts a set of target novel-view images. Given a single view of an object, unPIC comprises: (i) a geometry *prior* that infers

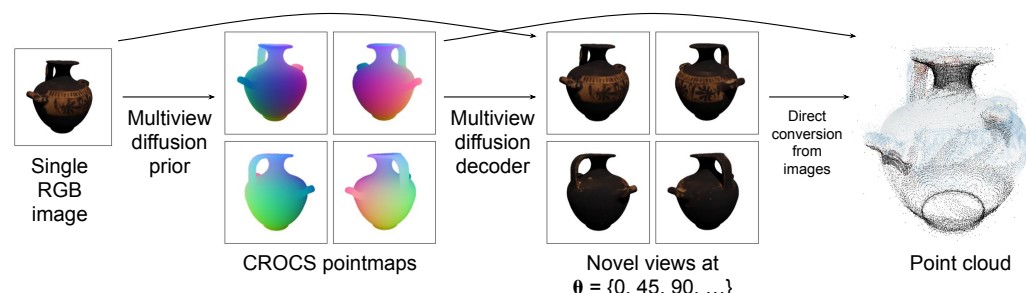

Figure 2: **The unPIC architecture.** A prior predicts multiview geometric features from a single source image. Our choice of features, CROCS, establishes point-to-point correspondence across views. The predicted CROCS images serve as a blueprint for the decoder module, which generates the fully textured target-view images, spinning the object around its vertical axis. We show 4 of 8 views here for clarity. All images are encoded as latents for diffusion. Once decoded, the output images can be directly converted to the vertices and vertex colors (respectively) of a 3D point cloud.

representations of the object's 3D geometry, and (ii) an appearance *decoder* that transforms all multiview geometry predictions into textured images of the subject. The two modules are trained independently. At inference time, we run the modules in a stacked fashion: the decoder takes the prior's outputs and the source image as inputs.

This specialization and separation of concerns has two advantages: one, it helps maximize the diversity of our samples at each stage. If the prior and decoder were trained end to end, the prior would need to output the most optimal representations for the decoder, and those could be texture-specific rather than geometric. A second advantage arises from the fact that the appearance decoder is only ever trained with ground-truth geometry: it adheres closely to the geometric conditioning.

## 3.2 MULTIVIEW DIFFUSION

Both the prior and decoder modules are implemented with identical multiview diffusion architectures, building on recent work (Wu et al., 2024a; Kong et al., 2024; Zheng & Vedaldi, 2024; Gao et al., 2024) that showed the benefit of predicting all K target views simultaneously. We found that tiling the views to form a single *superimage* works well; we tile them clockwise to ensure each view is adjacent to its closest neighbors in camera space (see Fig 2). Our U-Net architecture follows Gao et al. (2024), ensuring information exchange at two levels: between adjacent views using standard convolutions, and globally using self-attention across the superimage. Our implementation models K=8 views in a 2x4 grid. K=8 is sufficient to capture object geometry and appearance, and allows enough viewpoint overlap to teach the models to maintain multiview consistency, as further motivated in Appendix B.3. Since the target views are defined w.r.t. the source camera pose and remain fixed to produce a $360°$ spin, our model does not require explicit camera poses as input. The prior and decoder modules are trained from scratch, exclusively on Objaverse and ObjaverseXL assets.

We follow the Latent Diffusion approach (Rombach et al., 2021), using a VAE to encode the K views independently before tiling them to form a superimage. We fine-tune the Stable Diffusion 1.4 VAE separately on CROCS and RGB images from our training data under identical training conditions. While both VAEs start with similar PSNR scores ($\sim$37-38), the CROCS VAE's score goes up to 51 after fine-tuning; the RGB VAE's score improves slightly to 38. The CROCS VAE's boost in performance comes at a lower KL representational cost than the RGB VAE's (see Table 7 for detailed metrics). This is intriguing given that both VAEs started from the same pretrained checkpoint, which was never originally trained on CROCS images. We conclude that it is easier to reproduce CROCS pointmaps, which vary smoothly from pixel to pixel, than textured RGB images. This is yet another motivation for unPIC's approach of predicting geometry before appearance.

Our full framework can be described as follows: Let $z_k^g$ denote the geometry latent for the $k^{th}$ view (encoded from a CROCS pointmap) and $z_k^a$ denote the appearance latent (encoded from an RGB image). We assume both kinds of latents are standardized by their empirical mean & standard

deviation. They are then tiled to form the target superimages for geometry ($Z^g$) & appearance ($Z^a$):

$$Z^g = \text{tile-clockwise}(\{z_k^g\}_{k=1}^K) \quad \text{and} \quad Z^a = \text{tile-clockwise}(\{z_k^a\}_{k=1}^K)$$

Both the prior and decoder are conditioned on an input superimage, $X^a$, which is constructed from a masked set of K appearance latents. During training, we sample the index of a single source view $k^* \sim Uniform(1, \ldots, K)$. The input latent for this view, $x_{k^*}$, is set to its ground-truth appearance latent, $z_{k^*}$, whereas the remaining $K-1$ latents are replaced by 0. While we train on only one input view, randomizing its position teaches the model to process arbitrary input configurations at inference time. The resulting input latents, $\{x_k^a\}_{k=1}^K$, are tiled clockwise to form the final superimage:

$$X^a = \text{tile-clockwise}(\{x_k^a\}_{k=1}^K)$$

Finally, the unPIC inference pipeline can be written as:

$$\hat{Z}^g \sim f_{\text{prior}}(\cdot|X^a) \quad \text{and} \quad \hat{Z}^a \sim f_{\text{decoder}}(\cdot|X^a, \hat{Z}^g)$$

### 3.3 CROCS: CAMERA-RELATIVE OBJECT COORDINATES

Geometry underpins unPIC's hierarchical approach: we predict the 3D shape of an object before decoding its multiview appearance. To do so, we propose a dense pointmap representation called Camera-Relative Object Coordinates (CROCS). CROCS is a scale-free representation of geometry: the object/scene is first uniformly scaled to fit a 3D unit cube, so all spatial coordinates are in [0, 1]. These can therefore be interpreted as RGB colors and rendered as images, which maps naturally onto our multiview framework. While a single CROCS image serves as a dense annotation for a target novel view, a set of CROCS images taken together form the vertices of a 3D point cloud.

CROCS can be seen as a variation of Normalized Object Coordinates (Wang et al., 2019). Instead of orienting the color space w.r.t. the object's class-canonical pose, CROCS orients the scene geometry w.r.t. a given source camera. CROCS is also related to VGGT's view-invariant pointmaps (Wang et al., 2025), but those are scaled by a per-scene normalization factor, and not actually bounded.

**NOCS vs CROCS.** (i) While NOCS captures point-to-point correspondence between different instances of an object type, CROCS is intended to allow a pose-free treatment of the scene (see Figure 3). It acts as a data augmentation: a person's arm may be colored red or green or blue depending on the arm's position relative to the source camera. A model trained to predict CROCS will not understand the "default" pose of any object, but it will be able to learn the typical colors that can be observed from different sides of the CROCS cube. For example: the source-camera view is always colored white in the top-right-near corner and black in the bottom-left-far corner. This predictability is ideally suited for predicting novel views at regular intervals around the object, because each target view faces a different side/aspect of the CROCS cube, and consequently has a biased distribution of colors. (ii) Unlike NOCS, CROCS requires no segmentation or identification of the objects constituting the scene. CROCS treats a multi-object scene as one whole, allowing the model to focus on the spatial arrangement of different objects, rather than the poses of individual objects.

We provide a precise mathematical description of CROCS in Appendix A.1. We also address two geometric subtleties in working with CROCS in Appendix A.2—the first one is about renormalizing pointmaps each time the source camera is moved to ensure they are still bounded in [0, 1]. The second one explains why the CROCS cube remains axis-aligned, sitting squarely on the ground-plane, when the source camera is moved higher or lower (changing its elevation $\phi$).

## 4 EXPERIMENTS

We provide empirical results on our choice of representation in Sec 4.1; the quality of our novel-view synthesis in Sec 4.2; the accuracy of our 3D geometry in Sec 4.3, and finally an ablation of our hierarchical design in Sec 4.4.

### 4.1 IS CROCS A GOOD INTERMEDIATE REPRESENTATION?

To motivate geometric representations and the use of CROCS specifically, we run two preliminary experiments: one to assess what features are *predictive* of novel-view images, and another experiment to assess what features are *predictable* from a single source image.

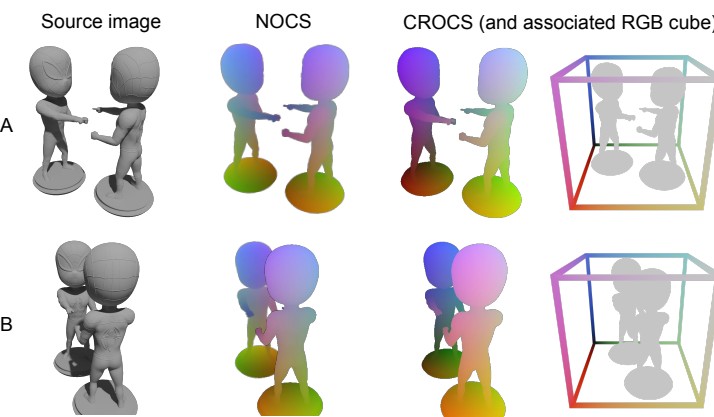

Figure 3: **Choice of pointmaps. Left:** source images derived by moving the camera. **NOCS:** objects are normalized and colored individually based on a class-canonical pose. The same semantic part receives the same color across instances, e.g., a right arm is always blue. **CROCS:** objects in the scene are normalized and colored collectively. Each pixel encodes normalized $(x, y, z)$ coordinates. The source camera sets the orientation of the coloring/coordinate system (shown as the RGB cube, fixed across examples). Any target viewpoint around the cube has a predictable color pattern.

Table 1: We analyze the effectiveness of CROCS for novel view synthesis (left) and its predictability from a single image (right) versus other geometric representations.

(a) **Does CROCS help predict novel views?** We predict a $90°$ object rotation using varied target image annotations. We report pixel MSE and stds. over 3 training runs.

| Annotation Type | MSE (1e-3) |
|---|---|
| Source image only | $19.042 \pm 0.722$ |
| + CLIP ViT L/16 | $6.059 \pm 0.448$ |
| + DINOv2 B/14 | $5.734 \pm 0.815$ |
| + Alpha mask | $8.169 \pm 0.392$ |
| + Depth map | $5.580 \pm 0.849$ |
| + NOCS | $5.318 \pm 0.070$ |
| + CROCS | $\mathbf{4.914} \pm 0.224$ |

(b) **Which multiview features are predictable from a single image?** We report the pointmap Mean Square Error (1e-3) at K=4 and K=8 target views.

| K | Annotation Type | MSE (1e-3) |
|---|---|---|
| 4 | NOCS | 11.94 |
|  | CROCS | **1.21** |
| 8 | NOCS | 15.82 |
|  | CROCS | **3.92** |

**Predictiveness.** We train a primitive version of unPIC's decoder module to predict a single novel view of an object at a fixed target rotation ($\theta = 90$ degrees). The model is conditioned on a source image, as well as a ground-truth annotation of the target novel view. The type of annotation is varied across training runs to assess the usefulness of different representations. The annotations range from geometric (like depth maps and CROCS), image-space maps (alpha masks), through feature maps (like DINOv2 (Oquab et al., 2024) and CLIP (Radford et al., 2021)). All annotations are treated as images—they are upscaled (if required) and concatenated to the input image before it is denoised. We run these experiments at 64x64 image size (to feasibly upscale the high-dimensional CLIP and DINOv2 features), and using pixel-based diffusion to avoid ambiguity about how to encode the annotations. Table 1a shows that CROCS is the most helpful for this task.

**Predictability.** Having shown that CROCS improves the decoder, we now ask: can CROCS itself be predicted reliably? We train primitive versions of the unPIC prior to predict either NOCS or CROCS from a single source image. We experiment with $K = 4$ or $K = 8$ target views. We compare two choices of representations: (i) NOCS images in a static reference frame, determined by the default creator-intended pose for the given asset; and (ii) CROCS, where we change the reference frame based on the source camera. We find a 4x increase in predictability from using CROCS (see Table 1b). With CROCS, each target pose has a biased distribution of colors, consistent across

Table 2: Summary of different approaches for Single-Image-to-3D. **P**: Probabilistic/Generative; **D**: Deterministic; **FF Recon**: Feed-Forward Reconstruction; **V-Gen → Opt**: View Generation followed by per-scene Optimization; **SVD**: Single-View Diffusion (run repeatedly per target viewpoint); **MVD**: Multi-View Diffusion (generates views jointly).

| Model | Output | Model Type | Generative Core | Intermediate 3D Rep. | Direct Geometric Supervision |
|-------|--------|-----------|-----------------|----------------------|------------------------------|
| LRM | D | FF Recon | - | Triplane → NeRF | No (Image loss only) |
| Free3D | P | V-Gen → Opt | MVD | Implicit (No 3D Rep) | No (Image loss only) |
| CAT3D | P | V-Gen → Opt | MVD | Implicit (3D Attention) | No (Image/Perceptual loss) |
| EscherNet | P | V-Gen → Opt | MVD | Implicit (MV Attention) | No (Image loss only) |
| One-2-3-45 | P | V-Gen → FF Recon | SVD | SDF (from Cost Vol.) | Depth |
| InstantMesh | P | V-Gen → FF Recon | MVD | Triplane → Mesh (FlexiCubes) | Depth & Normals |
| unPIC (ours) | P | Direct 3D + V-Gen | MVD (Hierarchical) | Pointmaps (CROCS) | Pointmaps (CROCS) |

objects. This makes the geometric feature space unambiguous, freeing up the model's probabilistic capacity for the ambiguous task of predicting the unseen object/scene geometry.

## 4.2 TASK 1: NOVEL VIEW SYNTHESIS

We compare unPIC against a diverse set of seminal and state-of-the-art baselines: CAT3D, Escher-Net, Free3D, OpenLRM, InstantMesh (Large), and One-2-3-45 XL. The key conceptual differences are summarized in Table 2 and detailed further in Appendix B.7.

**Test data.** We sample 512 test objects from 4 different datasets, including our held-out subset of Objaverse XL, and three out-of-distribution datasets: (i) Google Scanned Objects (Downs et al., 2022), (ii) Amazon Berkeley Objects (Collins et al., 2022), and (iii) the Digital Twin Catalog (Pan et al., 2023). The baselines may have trained on these test datasets—for instance, One-2-3-45 was trained on ObjaverseXL. For each test object, we sampled a random source and target view, and evaluated all models on the same pair.

**Metrics.** To measure novel-view prediction accuracy, we compute a number of image-quality metrics such as PSNR, FID (Heusel et al., 2017), LPIPS (Zhang et al., 2018), and SSIM (Wang et al., 2004). In addition, we focus on evaluating:

- 2D Geometric Accuracy: We compute the IoU (Intersection over Union) of the binarized mask of the predicted novel view w.r.t. the expected mask. This metric is invariant to lighting or texture.
- Multiview Consistency: Though we attempted to run MET3R (Asim et al., 2024) to measure multiview consistency, we found it inadequate for our use case, even for ground-truth images (see Appendix B.6). Instead, we compute CLIP (ViT L/14 336px) embeddings for K=8 generated images per object, including the source image. Then, we compute the mean of the (K x K) pairwise distances between the embeddings. This metric (abbreviated CLIP MV) helps probes the "internal" consistency of a set of generated images without comparing with ground-truth images.

**Results**. unPIC consistently outperforms the baselines on all novel-view metrics (Table 3). The IoU metric provides an early sign of its geometric accuracy. On the CLIP MV metric, it is noteworthy that the only method which outperforms unPIC consistently, InstantMesh, is also geometry-supervised.

## 4.3 TASK 2: 3D RECONSTRUCTION

We derive point clouds from unPIC by simply combining the multiview outputs. The CROCS images provide the vertices, while the RGB images provide the vertex colors; we only need to mask out the background pixels from the output images. We run two competing methods, Direct3D and InstantMesh (Large), on our test images for comparison. To ensure the Chamfer distances are comparable, we subsample 8192 points from every point cloud. We also provide the numbers reported by EscherNet, MeshLRM, and Direct3D; our Direct3D re-run produced a similar score as the original paper. Table 5 shows that unPIC outperforms all methods. See Figure 5 for a qualitative comparison.

## 4.4 ABLATING OUR HIERARCHICAL DESIGN

We train a version of unPIC without any geometry inputs or supervision, under identical training conditions. It is non-hierarchical as it takes a single RGB image and outputs the target novel views.

Table 3: **Novel-View Synthesis:** empirical comparison on 4 datasets.

| Dataset | Method | PSNR ↑ | IoU ↑ | FID ↓ | LPIPS ↓ | SSIM ↑ | CLIP MV (1e-4) ↓ |
|---|---|---|---|---|---|---|---|
| | CAT3D | 12 | 0.61 | 82 | 0.27 | 0.64 | 2.52 |
| | EscherNet | 13 | 0.63 | 87 | 0.24 | 0.68 | 3.13 |
| | InstantMesh | 13 | 0.62 | 76 | 0.25 | 0.67 | **1.89** |
| *Objaverse XL* | Free3D | 15 | 0.66 | 82 | 0.22 | 0.71 | 3.10 |
| | One-2-3-45 | 12 | 0.57 | 72 | 0.26 | 0.65 | 2.55 |
| | OpenLRM | 13 | 0.60 | 94 | 0.24 | 0.71 | 2.38 |
| | unPIC (ours) | **24** | **0.72** | **63** | **0.15** | **0.80** | 2.01 |
| | CAT3D | 19 | 0.64 | 87 | 0.26 | 0.72 | 2.98 |
| | EscherNet | 14 | 0.72 | 76 | 0.26 | 0.66 | 2.78 |
| | InstantMesh | 14 | 0.71 | 61 | 0.25 | 0.66 | **2.12** |
| *Google Scanned Objects* | Free3D | 14 | 0.70 | 93 | 0.25 | 0.68 | 3.63 |
| | One-2-3-45 | 13 | 0.64 | **60** | 0.26 | 0.65 | 2.74 |
| | OpenLRM | 14 | 0.66 | 109 | 0.26 | 0.69 | 2.93 |
| | unPIC (ours) | **23** | **0.77** | 62 | **0.17** | **0.77** | 2.33 |
| | CAT3D | 18 | 0.59 | 82 | 0.29 | 0.67 | 2.34 |
| | EscherNet | 11 | 0.65 | 73 | 0.30 | 0.59 | 2.35 |
| | InstantMesh | 12 | 0.67 | 60 | 0.27 | 0.62 | **1.54** |
| *Amazon Berkeley Objects* | Free3D | 12 | 0.69 | 73 | 0.27 | 0.63 | 2.92 |
| | One-2-3-45 | 11 | 0.61 | 50 | 0.28 | 0.61 | 2.22 |
| | OpenLRM | 11 | 0.57 | 122 | 0.30 | 0.65 | 2.55 |
| | unPIC (ours) | **25** | **0.88** | **40** | **0.15** | **0.78** | 1.74 |
| | CAT3D | 18 | 0.70 | 83 | 0.23 | 0.71 | 2.68 |
| | EscherNet | 14 | 0.74 | 75 | 0.23 | 0.68 | 2.37 |
| | Free3D | 14 | 0.76 | 87 | 0.22 | 0.70 | 2.90 |
| *Digital Twin Catalog* | InstantMesh | 14 | 0.74 | 58 | 0.22 | 0.68 | **1.51** |
| | One-2-3-45 | 14 | 0.74 | 53 | 0.21 | 0.69 | 1.98 |
| | OpenLRM | 14 | 0.70 | 98 | 0.22 | 0.72 | 2.40 |
| | unPIC (ours) | **25** | **0.90** | **46** | **0.13** | **0.82** | 1.55 |

Table 4: **Ablation:** we train a non-hierarchical version of unPIC without any geometry.

| Dataset | Method | PSNR ↑ | IoU ↑ | FID ↓ | LPIPS ↓ | SSIM ↑ | CLIP MV (1e-4) ↓ |
|---|---|---|---|---|---|---|---|
| *Objaverse XL* | unPIC ablation | 20 | 0.63 | 89 | 0.23 | 0.75 | 2.07 |
| *Google Scanned Objects* | unPIC ablation | 21 | 0.69 | 97 | 0.24 | 0.75 | 2.43 |
| *Amazon Berkeley Objects* | unPIC ablation | 19 | 0.61 | 84 | 0.28 | 0.70 | 1.84 |
| *Digital Twin Catalog* | unPIC ablation | 22 | 0.82 | 69 | 0.19 | 0.78 | 1.67 |

Table 5: **3D Geometric Accuracy:** we compute the bidirectional Chamfer distance (1e-3) ↓ to compare point cloud completeness and accuracy. All methods take a single image as input. GSO: Google Scanned Objects. DTC: Digital Twin Catalog.

| Method | GSO | DTC |
|---|---|---|
| EscherNet (Kong et al., 2024) | 31.4 | - |
| Direct3D (re-run) | 31.1 | 18.0 |
| Direct3D (Wu et al., 2024b) | 27.1 | - |
| Zero123++ & MeshLRM (Wei et al., 2024) | 15.7 | - |
| InstantMesh (re-run) | 6.83 | 6.62 |
| unPIC (ours) | **4.59** | **5.47** |

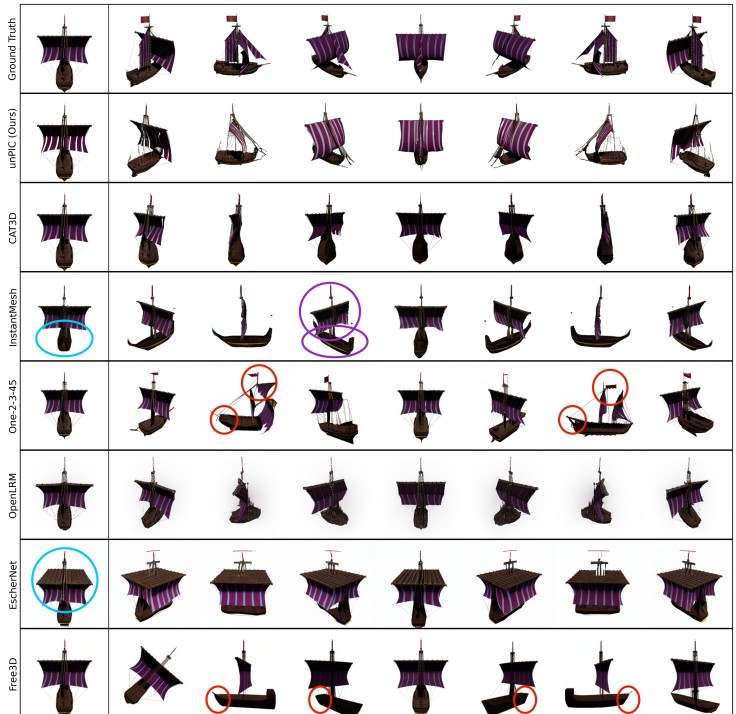

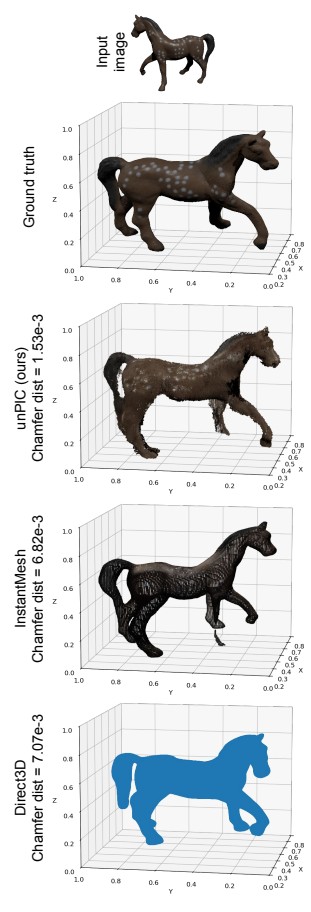

Figure 4: **Novel views, baselines vs unPIC.** The left column shows the source view, as output by each model. The models show different failure modes: (i) mutations of the original object (e.g., InstantMesh, EscherNet; annotated in blue); (ii) multi-view inconsistencies (e.g., One-2-3-45, Free3D; annotated in red); (iii) issues inferring the depth of the boat from the source image (e.g., CAT3D, OpenLRM); and (iv) pose inaccuracies in the novel views (e.g., InstantMesh gets the pose of the sail right, but the pose of the boat wrong; annotated in violet).

Figure 5: **Points clouds, baselines vs unPIC.**

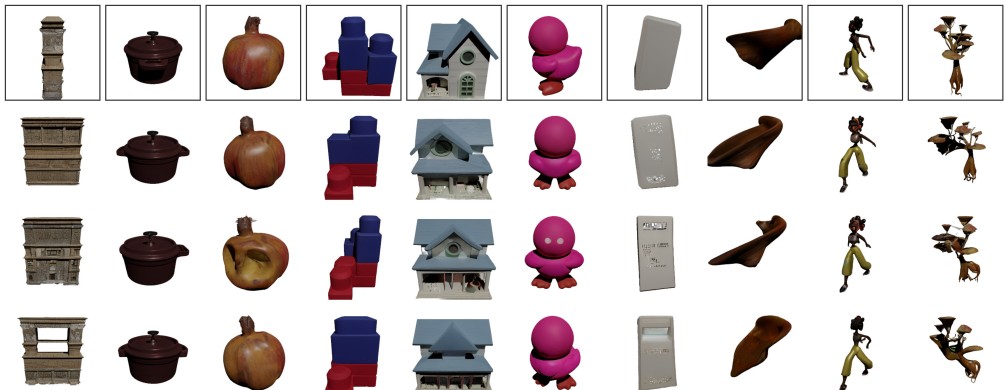

Figure 6: **Diversity of outputs.** Top row: various test images from DTC and ObjXL. Remaining rows: unPIC's output at a fixed 90° rotation across three RNG seeds, showing the range of outputs for the given conditioning.

In Table 4, we find that the ablated model performs much worse than the full version of unPIC; nevertheless, it is comparable to CAT3D, which is another geometry-free multiview synthesis method.

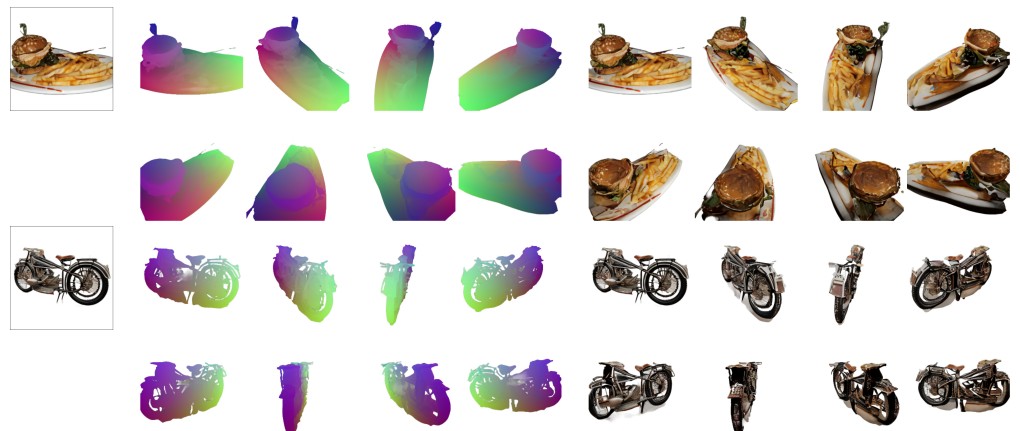

Figure 7: **unPIC on in-the-wild images.** Each pair of rows shows: the input image in the top-left box, predicted CROCS in the middle (2x4 grid), and predicted novel-view images to the right (2x4).

## 5 DISCUSSION

We demonstrate the advantage of unPIC's hierarchical approach by sampling multiple outputs while keeping the input image fixed in Figure 6. The output novel views remain plausibly constrained while exhibiting appropriate variety across random seeds. They show unPIC's ability to capture multimodal distributions in both geometry (varying structural interpretations of ambiguous inputs, e.g., the pomegranate or toy blocks) and appearance (texture variations, e.g., the white device).

Having never seen a real-world pixel during its training, unPIC nevertheless generalizes to in-the-wild images (see Figure 7). This is clear evidence that it is possible to learn real-world shape priors from synthetic assets alone. While our focus was on using these priors for image-to-3D generation, unPIC could also be used for vision tasks, e.g., monocular or stereo depth estimation and 3D annotation of 2D images.

**Limitations and failure cases.** (i) A salient shortcoming is that we do not model image backgrounds. This may be feasible while still using CROCS where the background geometry is uniform, and can be scaled to a unit cube (e.g., room-like scenes). (ii) unPIC rotates some objects in the image plane rather than spinning them around the vertical axis, likely because it misinterprets the source image as an overhead view of the object. Since CROCS does not account for the source camera elevation, we could potentially condition unPIC on the elevation to prevent ambiguity. We provide an example of this failure in Figure 15. (iii) Natural images containing faces or multiple people can be out-of-distribution, as unPIC was trained on synthetic assets lacking such cases. We hope that expanding our training data, e.g. by combining multiple assets in the same scene, could offer one avenue to ease this limitation. (iv) At present, unPIC is also limited by the expressiveness of the SD-VAE: for example, it cannot represent any text. A richer representation space could help get over this limitation.

**Conclusion.** We introduced the unPIC framework for image-to-3D generation. Our choice of intermediate features, CROCS, encodes point-to-point correspondence across views in a canonical reference frame. CROCS allows us to invert the usual "generate then reconstruct" paradigm, by producing 3D directly as part of the view generation process. With its hierarchical approach and explicit geometric supervision, unPIC outperforms numerous competing methods on novel-view synthesis metrics, multiview generation consistency, and 3D geometric accuracy. unPIC learns shape priors that generalize well beyond the digital assets it was trained on, paving the way for more geometry-driven novel view synthesis.

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

# A APPENDIX

## A.1 CROCS

We first provide a precise specification of CROCS.

**Normalization (similar to NOCS)** Let $\Omega \subset \mathbb{R}^3$ denote the surface of a 3D object or scene. For any point $\mathbf{p} = (x, y, z) \in \Omega$ in world coordinates, we first normalize its coordinates into NOCS, defined by the unit cube $[0, 1]^3$.

We determine the Axis-Aligned Bounding Box (AABB) of $\Omega$ using the minimum and maximum coordinates, $\mathbf{p}_{\min}$ and $\mathbf{p}_{\max}$. The geometric center of the AABB is $\mathbf{c}_{\Omega} = (\mathbf{p}_{\min} + \mathbf{p}_{\max})/2$. To ensure uniform scaling while preserving the aspect ratio, we calculate the maximum extent $L$ of the bounding box:

$$L = \|\mathbf{p}_{\max} - \mathbf{p}_{\min}\|_{\infty} = \max(x_{max} - x_{min}, y_{max} - y_{min}, z_{max} - z_{min}). \tag{1}$$

The NOCS coordinates $\mathbf{p}' \in [0, 1]^3$ are then obtained by scaling the object and centering it at $\mathbf{h} = (0.5, 0.5, 0.5)$:

$$\mathbf{p}' = \frac{1}{L}(\mathbf{p} - \mathbf{c}_{\Omega}) + \mathbf{h}. \tag{2}$$

**Camera-Relative Transformation** Let the position of a source camera be defined in spherical coordinates by $(\theta, \phi, r)$, corresponding to the azimuth, elevation, and radial distance, respectively. We assume the camera is always oriented to face the center of the object space.

To achieve camera-relative canonicalization, we rotate the normalized coordinates around the vertical axis (Z-axis) by the source camera's azimuth $\theta$. This aligns the coordinate system with the viewpoint of the source camera. The transformation is applied around the center of the unit cube $\mathbf{h}$. We define the intermediate rotated coordinates $\mathbf{p}''$ as:

$$\mathbf{p}'' = R_z(\theta)(\mathbf{p}' - \mathbf{h}) + \mathbf{h}, \tag{3}$$

where $R_z(\theta) \in SO(3)$ is the standard 3D rotation matrix around the Z-axis:

$$R_z(\theta) = \begin{pmatrix} \cos\theta & -\sin\theta & 0 \\ \sin\theta & \cos\theta & 0 \\ 0 & 0 & 1 \end{pmatrix}. \tag{4}$$

This transformation rotates the coordinates in the X-Y plane (corresponding to the Red and Green channels in the pointmap) while preserving the Z coordinate (Blue channel). Notably, this definition is deliberately invariant to the camera elevation $\phi$ and distance $r$, a choice further elaborated in Appendix A.2.2.

**Rescaling** Since the object was initially scaled to fit an axis-aligned unit cube (Eq. 2), the rotation in Eq. (3) may cause the coordinates $\mathbf{p}''$ to exceed the $[0, 1]^3$ bounds (illustrated in Appendix A.2.1). A final rescaling step is necessary to ensure the coordinates remain constrained within the unit cube.

To ensure the normalization is robust and predictable across different azimuth angles, we incorporate the theoretical bounds of the transformation. We define the theoretical range $L_{th}(\theta)$ and the theoretical minimum coordinate $m_{th}(\theta)$ achievable when rotating a unit cube by $\theta$ around its center:

$$L_{th}(\theta) = |\cos\theta| + |\sin\theta|, \tag{5}$$

$$m_{th}(\theta) = 0.5 - \frac{1}{2}L_{th}(\theta). \tag{6}$$

Let $\Omega''$ be the set of all rotated coordinates for the object. We calculate the observed minimum $m_{obs}$ and maximum $M_{obs}$ values across all points and all dimensions of $\Omega''$. The effective minimum shift $m_{eff}$ and the effective range $L_{eff}$ are calculated by clamping the observed values with the theoretical bounds. This ensures that the normalization is robust and consistent with the geometric constraints of the rotation:

$$m_{eff} = \max(m_{obs}, m_{th}(\theta)), \tag{7}$$

$$L_{eff} = \text{clip}(M_{obs} - m_{eff}, \epsilon, L_{th}(\theta)), \tag{8}$$

where $\epsilon$ is a small positive constant (e.g., $10^{-6}$) to prevent division by zero.

The final CROCS coordinates $\mathbf{p}_{\text{CROCS}} \in [0,1]^3$ are given by the uniformly rescaled coordinates:

$$\mathbf{p}_{\text{CROCS}} = \frac{\mathbf{p}'' - m_{eff}}{L_{eff}}. \tag{9}$$

This procedure guarantees that the object remains tightly bounded by the unit cube in the camera-relative coordinate space.

## A.2 GEOMETRIC SUBTLETIES

### A.2.1 CROCS RESCALING

Recall that we pre-render multiview NOCS images using a fixed reference frame. This reference frame can be labeled using the source-view camera position $(\theta, \phi, r) = (0, 0, 1)$ that captures the object in its default creator-intended pose at a default camera distance. At training time, we would like to set the source camera to arbitrary locations to ensure a rich training distribution. To change the reference frame from $\theta = 0$ to a new source-camera location $\theta'$ (we will treat the other parameters in Appendix A.2.2), we simply rotate the *ground-plane axes* of all pre-rendered NOCS maps using the SO2 rotation matrix $[[\cos\theta', -\sin\theta'], [\sin\theta', \cos\theta']]$. The ground-plane axes are the Red and Green channels of the NOCS maps in our case.

Our on-the-fly NOCS-to-CROCS canonicalization comes with a challenge, arising from the fact that the object was scaled to fit the NOCS cube at $\theta = 0$. When we rotate the RGB reference cube to follow the primary camera to $\theta'$, then the object may no longer be bounded by the rotated cube. See Figure 8 where we visualize this issue for a fixed object scaled to fit the reference cube at $\theta = 0$.

In the upper row of Figure 8a, we get some CROCS values outside the range $[0,1]$, because the object exceeds the boundaries of the $[0,1]^2$ CROCS reference square. Some values of $\theta'$ such as 45, 135, 215, and 305 degrees are affected the most, because they lead to the greatest object overhang. This is not ideal—the scale of CROCS values which the model needs to predict depends on the reference frame. Since the reference frame (i.e., source camera position) is not actually supplied to the model, the model can only learn to predict the variable scale by overfitting. We aim to avoid any dependence on the reference frame. To this end, we introduce a CROCS rescaling operation to eliminate the dependence (Figure 8a, lower row).

Taking a concrete example, Figure 8b shows a cube-like object from different angles, along with the pre-rendered NOCS images (second row), and canonicalized CROCS maps without and with rescaling (third and fourth rows). Here are two observations from this figure: **1)** The front-facing edge of the cube, visible at $\theta' = 45$ as yellow in the second row, disappears in the third row because the CROCS values exceed the $[0,1]$ range (and the color map is defined on the same range). If we rescale the values in the third row, we see the edge reappears in the fourth row. **2)** The maximum overhang (in the third row) occurs at $\theta' = 45$, where the range of observed $y$ values expands to $[-0.2, 1.2]$. The length of this expanded interval is nearly $\sqrt{2} \approx 1.41$, which can also be deduced geometrically—it equals the length of the diagonal of the fixed square in Figure 8a.

Geometrically, we expect no object overhang at all if a given object is bounded by a sphere of diameter 1 (a tighter bound than the unit cube). We believe a lot of objects fall in this category. For this reason, our model does reasonably well even if we train it to predict CROCS without rescaling. But we observed a noticeable gain in the prior's performance on CROCS with rescaling. The latter setting removes the reference-frame-dependent scale that the model needs to predict in some cases (objects like a cube).

Note that for a given object, the rescaling needs to be performed across all multiview pointmaps (K=8 target views in our case) simultaneously, since any given view only shows a particular facet of the object. (A single pointmap may entirely miss the variation along its depth component.) Our rescaling is reminiscent of the pairwise pointmap renormalization performed by DUSt3R (see Eq. 3 in Wang et al. (2024a)). All the results in this work use CROCS with rescaling.

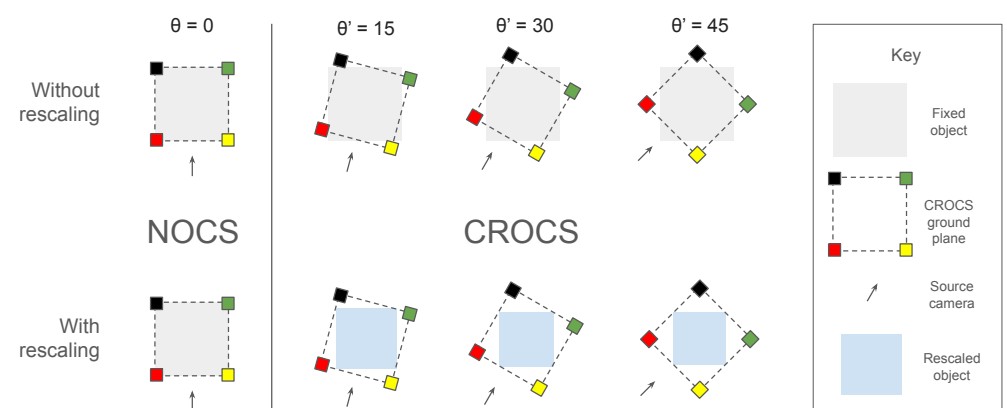

(a) **The general case** (simplified to 2D). **Top row:** An object that fits the NOCS reference square exactly at $\theta = 0$ is no longer bounded by the CROCS reference square at $\theta' \in \{15, 30, 45\}$. Due to object overhang, some CROCS values lie outside $[0, 1]$. The extent of the overhang depends on $\theta'$. **Bottom row:** We rescale CROCS based on the observed range of multiview values for any $\theta'$. This ensures the object is once again bounded tightly by $[0, 1]^2$.

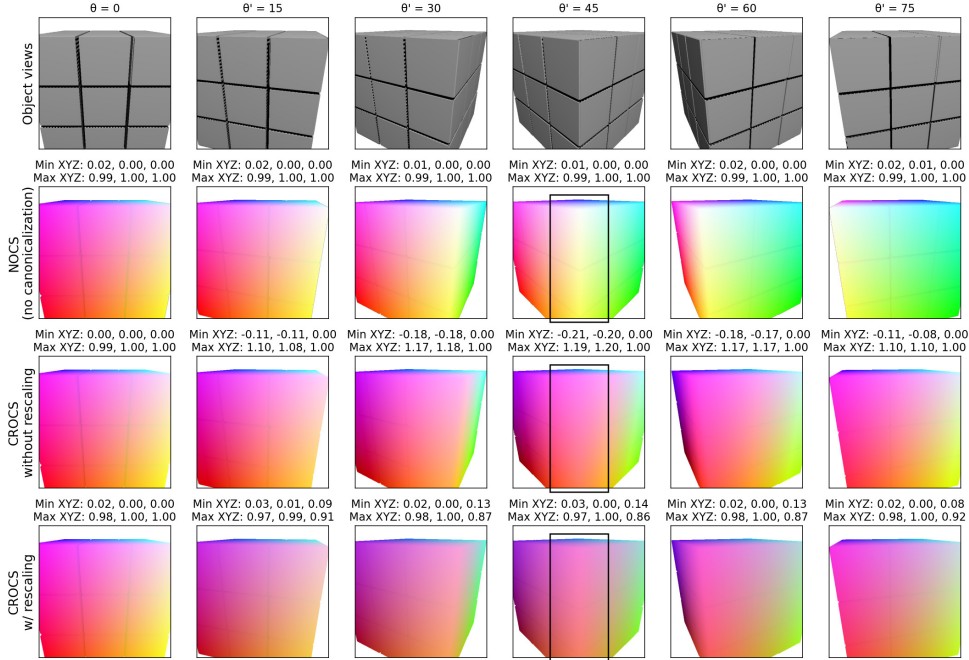

(b) **A concrete example in 3D.** We show the minimum and maximum NOCS/CROCS values observed above each pointmap image. At $\theta' = 45$, we see that the front-facing edge of the cube appears squashed without rescaling, because its X and Y values lie outside the range $[0, 1]$.

Figure 8: **CROCS rescaling.** We examine the effect of rotating the RGB reference cube used to paint the object's surface, following the source camera as it moves around a fixed object. $\theta$ (the azimuthal angle) denotes the default camera position, while $\theta'$ denotes a new position. We consider the largest possible objects—a square in 2D or cube in 3D.

### A.2.2 NO CANONICALIZATION FOR CAMERA ELEVATION OR DISTANCE

In Appendix A.2.1 we discussed how to canonicalize NOCS when switching the source camera from $\theta$ to $\theta'$. Here, we discuss the remaining camera extrinsic parameters. Since all cameras are oriented

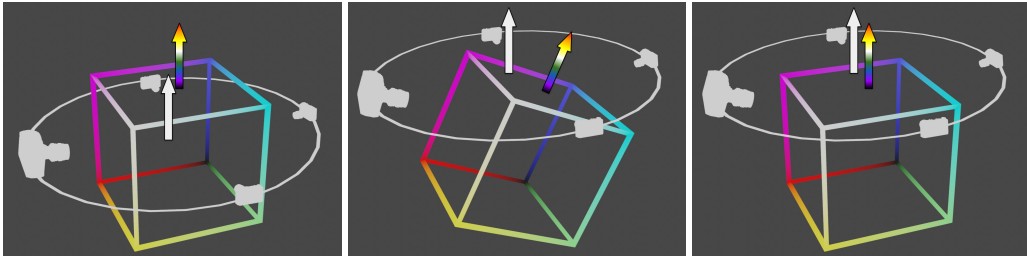

Figure 9: **CROCS when varying the camera elevation angle** $\phi$. As in Fig. 3, the large camera denotes the source view; the smaller cameras denote 3 of 7 novel views. The white arrow denotes the normal direction w.r.t. the camera plane (drawn as a white circle connecting the cameras). The colored arrow denotes the normal of the CROCS color cube. **Left:** at the default angle $\phi = 0$, each camera is pointed at a particular face of the RGB color cube, and has a consistent color distribution across examples. **Middle:** when the source camera is raised to $\phi'$, we could hypothetically rotate the color cube by the same degree to ensure the source camera still points at the same face of the cube. The issue is—tilting the color cube toward the source camera tilts it away from the opposite camera, and also rotates the colors seen by the remaining target cameras (by $\phi'$ and $-\phi'$ in their respective image planes). The color shifts experienced by the four cameras are clearly inconsistent. **Right:** If we choose not to tilt the cube through $\phi'$, all cameras undergo a small and consistent shift in the distribution of colors toward the up-axis color (blue). If the model can infer $\phi'$, it can also predict the shift. We follow the *Right* approach.

to face the center of the object, we do not need to consider their rotations. In fact, we do not need to canonicalize for changes in the camera distance $r$. It only serves as a zoom parameter to augment the training data.

The only remaining parameter is the elevation angle $\phi$, which determines the source-camera height. Recall that the target cameras are also placed at the same camera height. In fact, $\phi$ determines the difficulty of the multiview prediction problem. For instance, at $\phi = 90$ (a straight-down view of the object), the source and target camera locations converge to the same point—the apex of the hemisphere. In this case, the prediction task is reduced to rotating the object in the image plane.

Say we raise/lower the source camera from $\phi$ to $\phi'$ (Figure 9). This would reveal more/less of the object's upper/lower surface, and hence more/less of the NOCS color representing the up-axis (Blue in our case). We could potentially correct for this by tilting the reference cube so the source camera is pointed at the same cube face as before at $\phi$. This would help maintain the same color bias for the source camera. However, the target cameras would no longer point at the same cube faces respectively—rather, their distribution of colors would shift (toward the up-axis/away from it in the case of the opposite target camera) or rotate (in the image plane for other target cameras). Consequently, the distribution shifts in the CROCS colors would be inconsistent across target cameras.

Based on this geometrical observation, we choose not to canonicalize for changes in the camera elevation $\phi'$. This has the effect of leaving it to the model to infer the camera elevation from the source image (implicitly in contrast to One-2-3-45 (Liu et al., 2024)). It differs from our treatment of $\theta'$, which leaves the model oblivious to the azimuthal rotation of the cameras.

Our decision not to canonicalize for $\phi$ is forced by the design choice of our target camera poses, which follow the camera height of the source camera. An alternative choice would be to tilt the camera plane and the CROCS cube together, leaving the cameras at different heights in Figure 9-Middle[1]. Our choice of target poses is based on how humans perceive and reason about objects. We tend to see side or top-down views of *level* objects. On mental rotation tasks (e.g., when asked what an object looks like from "behind"), we tend to imagine a rotation of the object either in the image plane, or in depth about the world's vertical axis (Shepard & Metzler, 1971; Vandenberg & Kuse, 1978). We find it harder to imagine a potentially bottom-facing view. In addition to aligning with

---

[1]This alternative is equivalent to using a stationary multiview camera rig, but orienting the object randomly before taking pictures, as in CAT3D (Gao et al., 2024), likely LRM (Hong et al., 2024), and datasets like YCB (Calli et al., 2015). Changing the object's orientation decouples the object's up-axis from the world's up-axis. It can leave the object in unnatural, physically unstable poses.

human perception, modeling object rotations at typical views/poses (e.g., top-down or side angles) could also be more useful for downstream applications.

# B  TRAINING AND EVALUATION DETAILS

## B.1  DATASET

We use the following sets of assets from Objaverse: 1) the LVIS subset expanded to 83k examples from the same categories. We use object type labels predicted with high confidence from (Kabra et al., 2024) to expand the LVIS subset. 2) The KIUI subset [2] comprising 101k additional assets. 3) From the Objaverse-XL alignment subset (Deitke et al., 2024), we used a combination of GLB, OBJ, and FBX assets after filtering out certain (a) terrains and HDRI environment maps, (b) layouts and rooms, and (c) textureless FBX assets that rendered as pink. In total, we used 620k assets (Objaverse and Objaverse-XL) for training. In addition, we held out 50k assets randomly sampled from Objaverse-XL after the filtering step.

To render NOCS images, we adapted a script from BlenderProc[3] (Denninger et al., 2023) which creates a special NOCS material for the surface of a given object. We also use their Blender settings (the CYCLES engine with 1 diffuse bounce, 0 glossy bounces, and 0 ambient occlusion bounces) for rendering NOCS. We export them in the EXR format to ensure linear-light values, preserving the continuity of NOCS values on the object's surface.

## B.2  MODEL HYPERPARAMETERS

We model images at $256 \times 256$ resolution. When encoded to latents, the dimensionality reduces to $32 \times 32 \times 4$. Hence, each superimage comprising K=8 views has dimensionality $64 \times 128 \times 4$.

We train the diffusion models using a standard cosine noise schedule, and $t \in [0, 1000]$. The denoiser predicts $\epsilon$, the noise added to the superimage $Z_t$ at time $t$. We use the standard diffusion loss, $L_\epsilon = \mathbb{E}_{t,\epsilon,Z_0}||\epsilon - f_\theta(Z_t, t)^2||.$

We train the prior and decoder modules with a conditioning dropout probability of 0.08 on the source image. We do not use any dropout on the CROCS images fed to the decoder to ensure it adheres to the geometry. At test time, we use a fixed classifier-free guidance weight of 2.0 on the source image for both modules. We use an exponential moving average of the model parameters for evaluation, although this is not essential.

The unPIC base model has a total of 1.1B parameters across the prior (470M), decoder (470M), and two VAEs (84M each). It takes about 1min 13s to run all model components, with classifier-free guidance over 1000 denoising steps, on an A100 GPU. Note that we did not attempt to improve the runtime, as that was not the core aim of this work. Techniques such as step distillation, or recent advances in Optimal Flow Matching (Kornilov et al., 2024) which facilitate straight trajectories for fast denoising, would be readily applicable to unPIC.

## B.3  CHOICE OF K=8 VIEWS

To motivate why $K = 8$ views are sufficient to produce a good high-level 3D reconstruction, we take a larger number of ground-truth views ($K = 24$), and look at the pairwise similarity between them. This can be computed using CLIP embeddings (like our CLIP-MV metric). See Figure 10 for a heatmap of pairwise similarity scores.

The black crosses that occur in the heatmap show the frequency at which rotated views become dissimilar to each other. We see 6 black crosses on either side of the diagonal of the matrix. This number estimates how many views are sufficient to summarize the full 360-degree rotation. Hence, by generating 7 novel views ($K = 8$) for any input image, we can capture 3D structure at sufficient granularity for reconstruction.

Note that this analysis does not take into account top and bottom views of the object.

---

[2]https://github.com/ashawkey/objaverse_filter
[3]https://github.com/DLR-RM/BlenderProc/

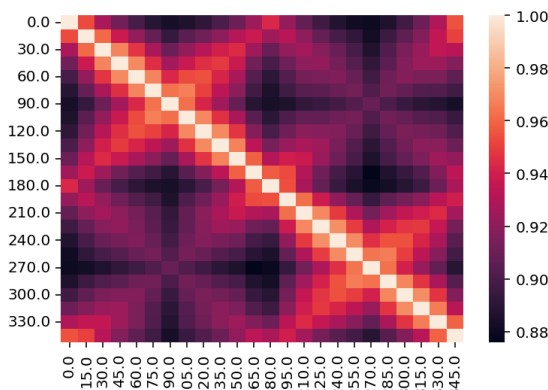

Figure 10: **Pairwise cosine similarities of CLIP embeddings for rotations of a given object**, averaged across a subset of ObjaverseXL. The x- and y-axes both show 24 views across a 360-degree rotation.

### B.4 U-NET ARCHITECTURE

The denoising network follows a standard design with a downsampling encoder path, a bottleneck middle path, and an upsampling decoder path with skip connections. The exact architecture is as follows, while the hyperparameters are summarized in Table 6:

- Input: A noisy latent superimage $X$ and a time-step embedding $t_{emb}$ derived from the diffusion time $t$. Any conditioning superimages (e.g., CROCS latents, $Z^g$) are concatenated with $X$ along the channel axis.

- Downsampling Stack: This path consists of five levels. The superimage is downsampled $2\times$ at each step. In total, this reduces the superimage from $64\times128$ down to a $2\times4$ feature map, where each of the 8 views is represented by a $1\times1$ spatial embedding. Each downsampling level contains three Resnet Blocks, and the number of channels is given by `embed_dim * channel_mult[i]`, where `i` is the downsampling level. A Multi-Head Attention Block is applied at all spatial resolutions below and including $32\times64$.

- Middle Stack: This path consists of three Resnet Blocks with Multi-Head Attention Blocks in between them to process the lowest-resolution feature map.

- Upsampling Stack: This path mirrors the downsampling path. At each level, the feature map is upsampled, concatenated with the corresponding skip connection from the downsampling path, and passed through three Resnet Blocks. A Multi-Head Attention Block is also applied at specified resolutions.

- Output: The final feature map is processed through a normalization layer, a Swish activation, and a final $3\times3$ convolution to produce the denoised latent superimage of the same shape as the input $X$.

Core Components:

- Resnet Block: A standard residual block containing two $3\times3$ convolutional layers, Group Normalization, and Swish activations. The time-step embedding $t_{emb}$ is projected and used to perform a feature-wise affine transformation (scale and shift) on the feature map after the first normalization layer.

- Multi-Head Attention Block: A standard multi-head self-attention mechanism applied over the spatial dimensions of the feature map. This allows every location in the feature map to integrate information from all other locations, enabling global communication across the different views tiled in the superimage.

Table 6: U-Net Hyperparameters

| Parameter | Value |
| --- | --- |
| embed_dim | 256 |
| channel_mult | (1, 2, 2, 3, 3) |
| num_res_blocks | 3 |
| per_head_channels | 64 |
| unet_dropout | 0.0 |
| norm_type | 'Group Norm' |

Table 7: **VAE metrics** on ObjaverseXL before and after fine-tuning on (a) CROCS pointmaps and (b) RGB images.

|  |  | RGB VAE | CROCS VAE |
| --- | --- | --- | --- |
| Before Fine-Tuning | PSNR | 37 | 38 |
|  | FID | 62 | 16 |
|  | KL | 28005 | 20823 |
| After Fine-Tuning | PSNR | 38 | 51 |
|  | FID | 16 | 2 |
|  | KL | 87496 | 35936 |

Google Scanned Objects
MEt3R score of pair: 0.3374

Amazon Berkeley Objects
MEt3R score of pair: 0.3907

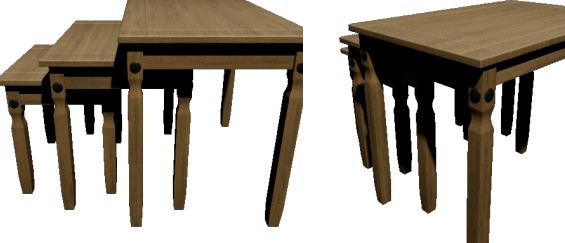

Figure 11: **MET3R as a metric for multiview consistency** produced values in excess of those expected from random noise, even on ground-truth images. The 45-degree rotation in each case is too great to facilitate a dense 3D reconstruction.

## B.5 VAE FINE-TUNING METRICS

We fine-tuned the Stable Diffusion 1.4 VAE, on CROCS and RGB images separately, using a combination of losses: Reconstruction, LPIPS, KL-Divergence, and the GAN loss. See Table 7 showing the improvement in each VAE's metrics after fine-tuning. The CROCS VAE benefits significantly, and outperforms the RGB VAE in terms of accuracy.

## B.6 ASSESSING MULTIVIEW CONSISTENCY USING MET3R

To assess multiview consistency, we attempted to run MET3R, which is a pose-free way to evaluate the consistency of generated novel views or frames of a generated video. Using ground-truth images, we found that even a 45-degree rotation of an object is too significant for MET3R. The metric produced values (see Fig 11) exceeding the expected value of the metric on random noise (around 0.3). We ran with the standard settings, using the MAST3R backbone, DINO16 features, FeatUp upsampling, and the cosine distance.

## B.7 BASELINES

For single-image-to-multiview synthesis, we compare unPIC with the following baselines:

1. **CAT3D** (512x512): a SoTA-quality, geometry-free diffusion model trained using masked modeling of views on diverse datasets. While the original model is closed source, we received a Colab and checkpoint from the authors. The model uses a coarse view-sampling strategy to generate 7 anchor views given 1 source image. Subsequently, it conditions on the anchor views to generate a finer set of views using the same model. We focus on the coarse stage for our evaluation.

2. **EscherNet** (256x256): a geometry-free diffusion model equipped with a specialized camera encoding (CaPE), allowing a flexible number of input images and target views. It uses the pretrained StableDiffusion v1.5 model as its latent diffusion backbone.

3. **Free3D** (256x256): a geometry-free multiview diffusion model that uses Plücker rays to encode target camera poses and modulate the diffusion latents. It is trained on Objaverse only.

4. **InstantMesh** (512x512): a geometry-aware, feed-forward framework that generates a 3D mesh from a single image. We use the Large variant of the model. InstantMesh first employs a multi-view diffusion model, Zero123++, to produce a set of 3D-consistent views from the input. These views are then processed by a transformer-based sparse-view Large Reconstruction Model (LRM) that directly outputs a mesh. The model was trained with direct geometric supervision (depths and normals) on the mesh representation using a differentiable iso-surface extraction module (FlexiCubes). We use the frames rendered from the 3D mesh for our evaluation.

5. **One-2-3-45** (256x256): a SoTA version of Zero-1-to-3 (Liu et al., 2023). Its diffusion-based image prior takes camera rotation and translation (R and T) parameters to transform a given image. Each novel view is sampled individually, leaving the outputs prone to multiview inconsistencies. We focus on evaluating the geometry-free stage, feeding R and T parameters to match our target poses, but using the model's own estimation of the camera height.

6. **OpenLRM** (288x288): a geometry-aware but deterministic method based on the original LRM (Hong et al., 2024). It uses a Transformer to convert DINO features into implicit triplane representations of 3D shape. The triplane tokens are spatially indexed/queried to parameterize a NeRF model of the scene, mitigating view consistency issues and allowing image-generation at arbitrary camera poses.

## C ADDITIONAL RESULTS

**Task 1: Novel-view synthesis.** We provide additional multiview comparisons between unPIC and the baselines in Figure 12. We show five examples at eight target views each.

**Task 2: 3D reconstruction.** We visualize our point clouds and compare them with two baselines in Figure 13. We also show meshes obtained by applying a surface-reconstruction algorithm (Screened Poisson (Kazhdan & Hoppe, 2013)) on our point clouds. We show 40 examples at one novel view each.

**In-the-wild generalization.** We apply unPIC to arbitrary real-world images in Figure 14. We also cover failure cases in Figure 15.

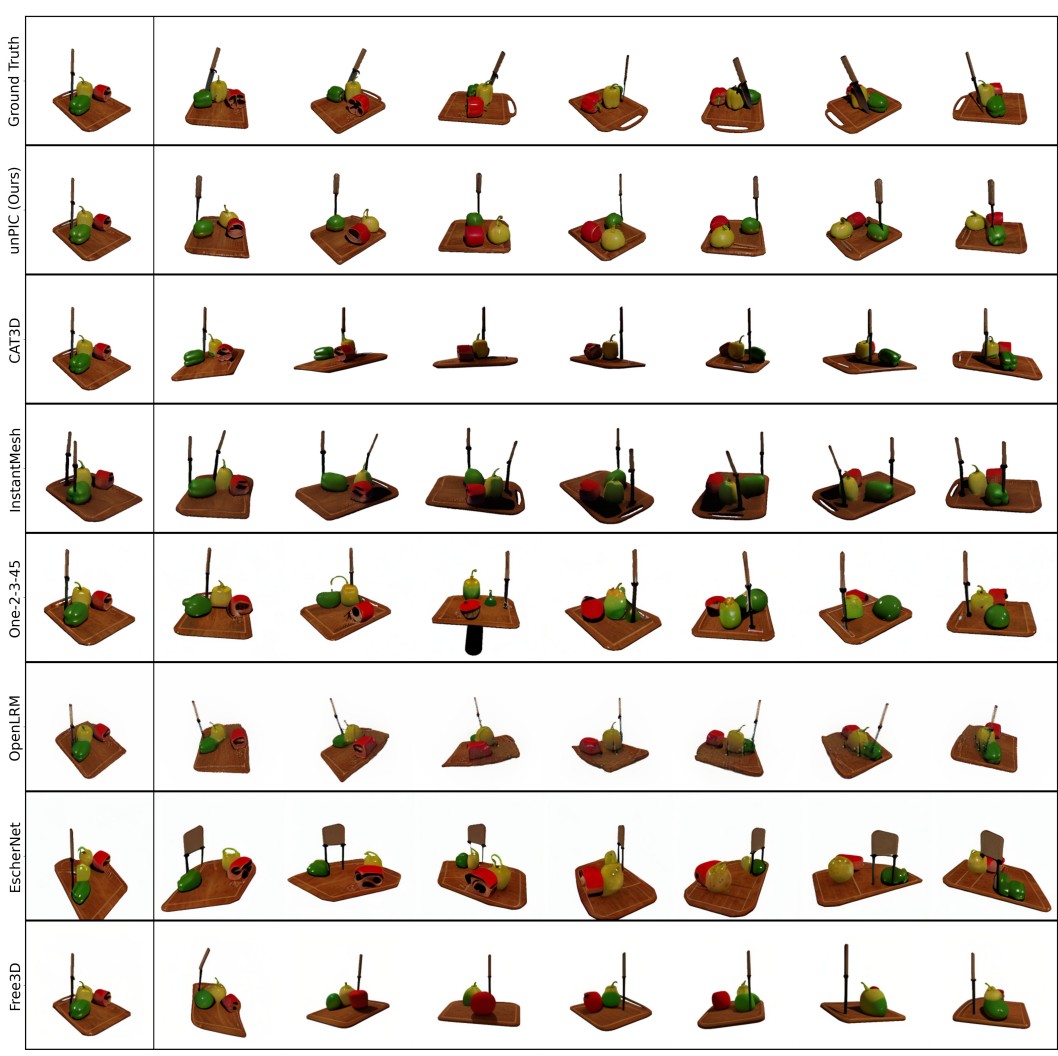

Figure 12: **Novel views from baselines versus our method** (figure continues on next page). We show an example from our ObjaverseXL test set.

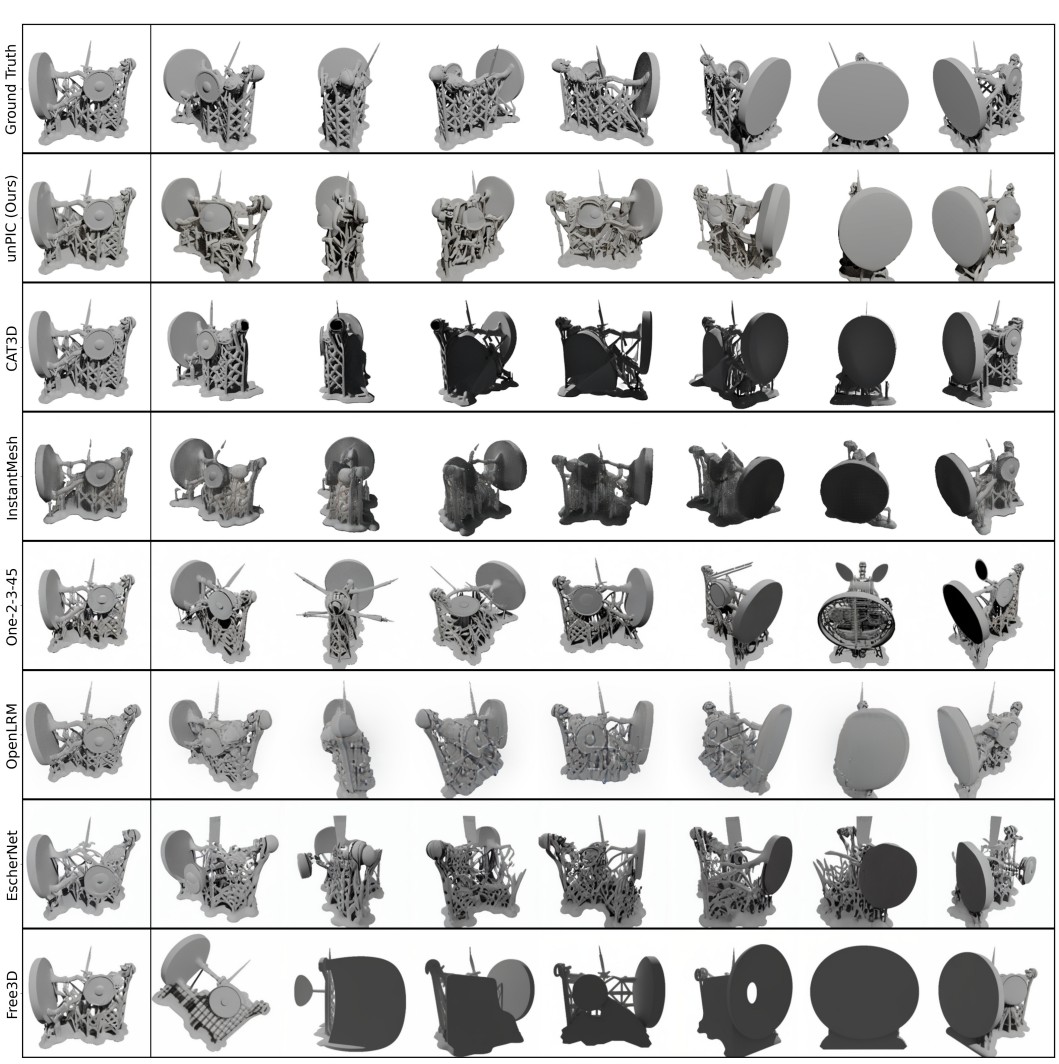

Figure 12: **Novel views from baselines versus our method** (figure continues on next page). We show an example from our ObjaverseXL test set.

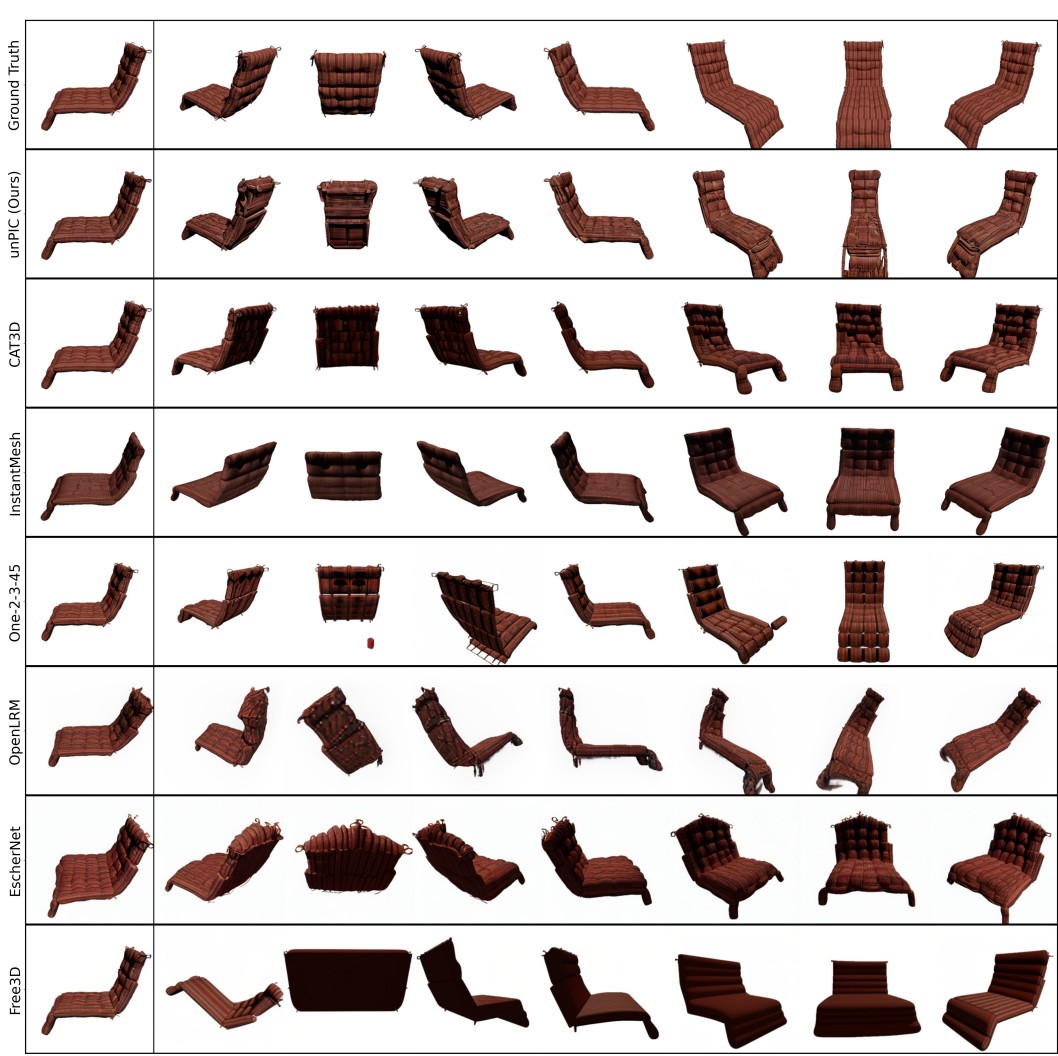

Figure 12: **Novel views from baselines versus our method** (figure continues on next page). We show an example from our Amazon Berkeley Objects test set.

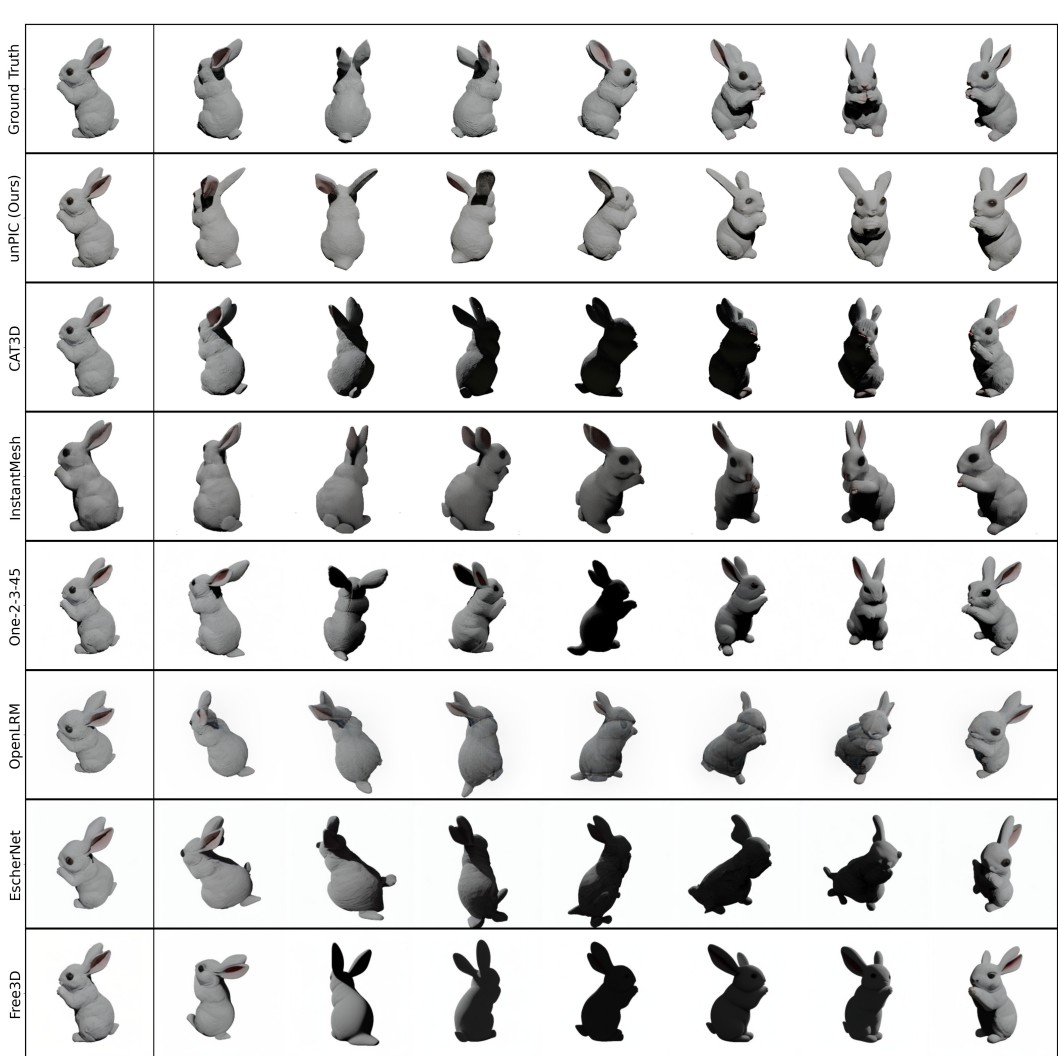

Figure 12: **Novel views from baselines versus our method** (figure continues on next page). We show an example from our Digital Twin Catalog test set.

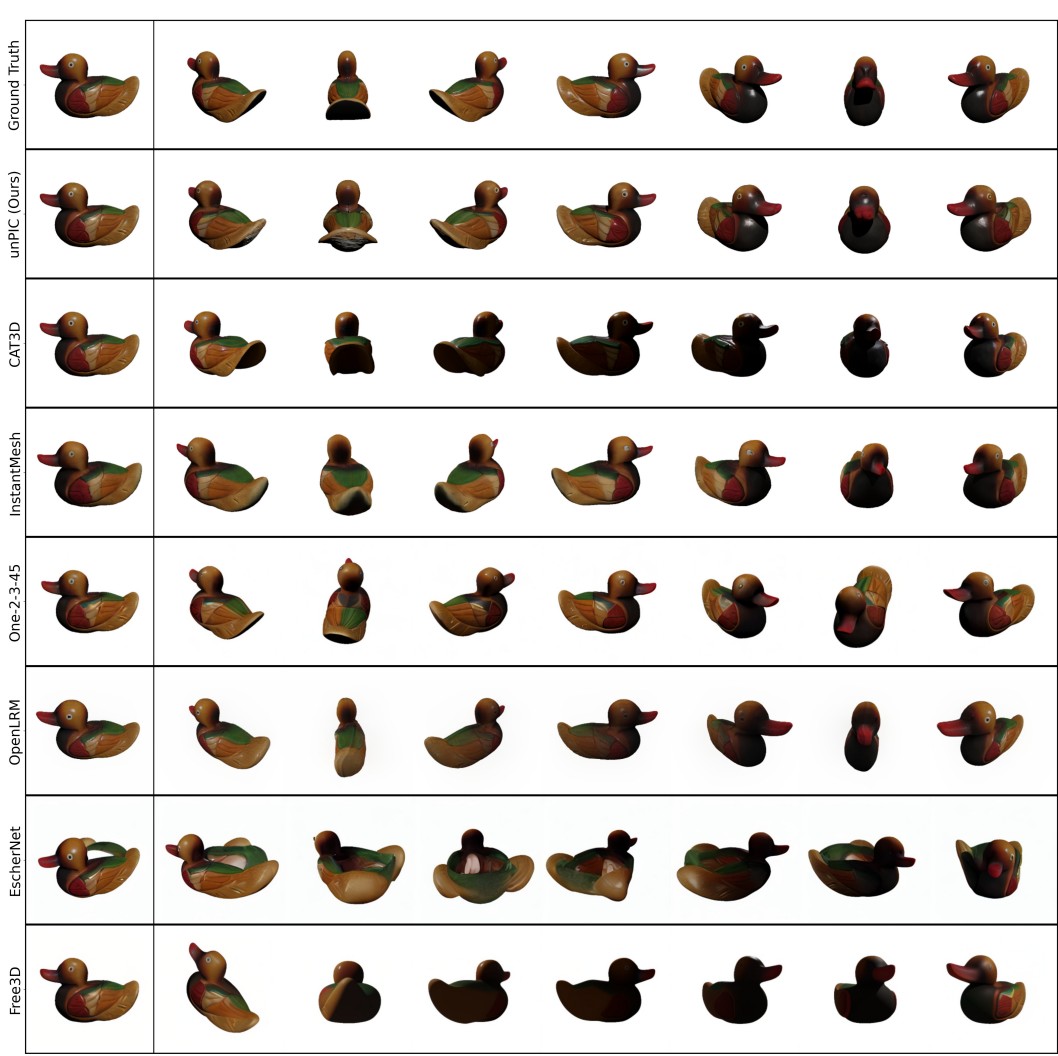

Figure 12: **Novel views from baselines versus our method** (continued). We show an example from our Digital Twin Catalog test set.

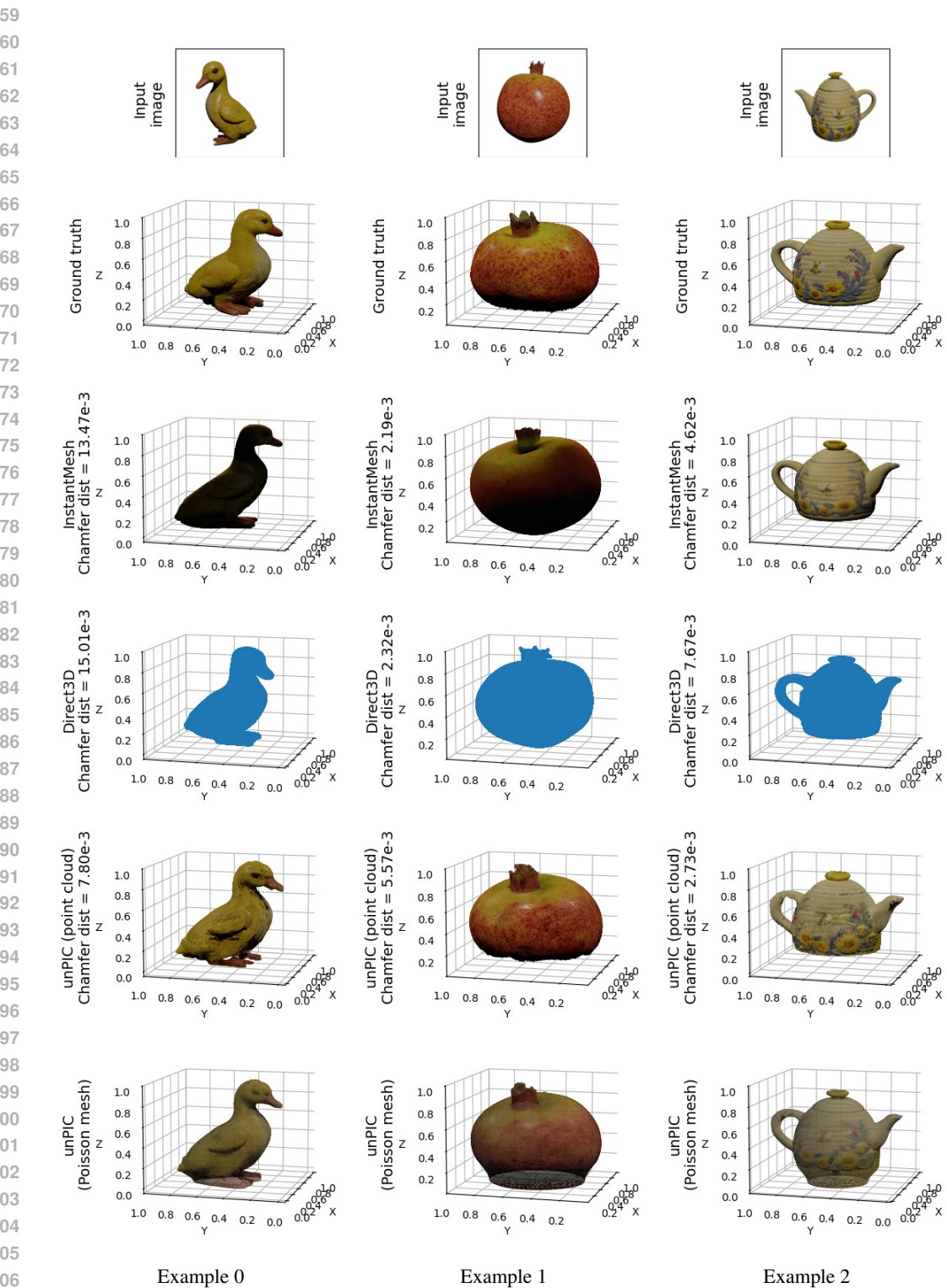

Example 0   Example 1   Example 2

Figure 13: **3D reconstruction results from unPIC vs baseline methods (Part 1 of 14).** We show a 195° rotation of the object relative to the input image. The bottom row (unPIC's reconstructed mesh) is visualized with transparency to show the full 3D shape. (Figure continues on next page.)

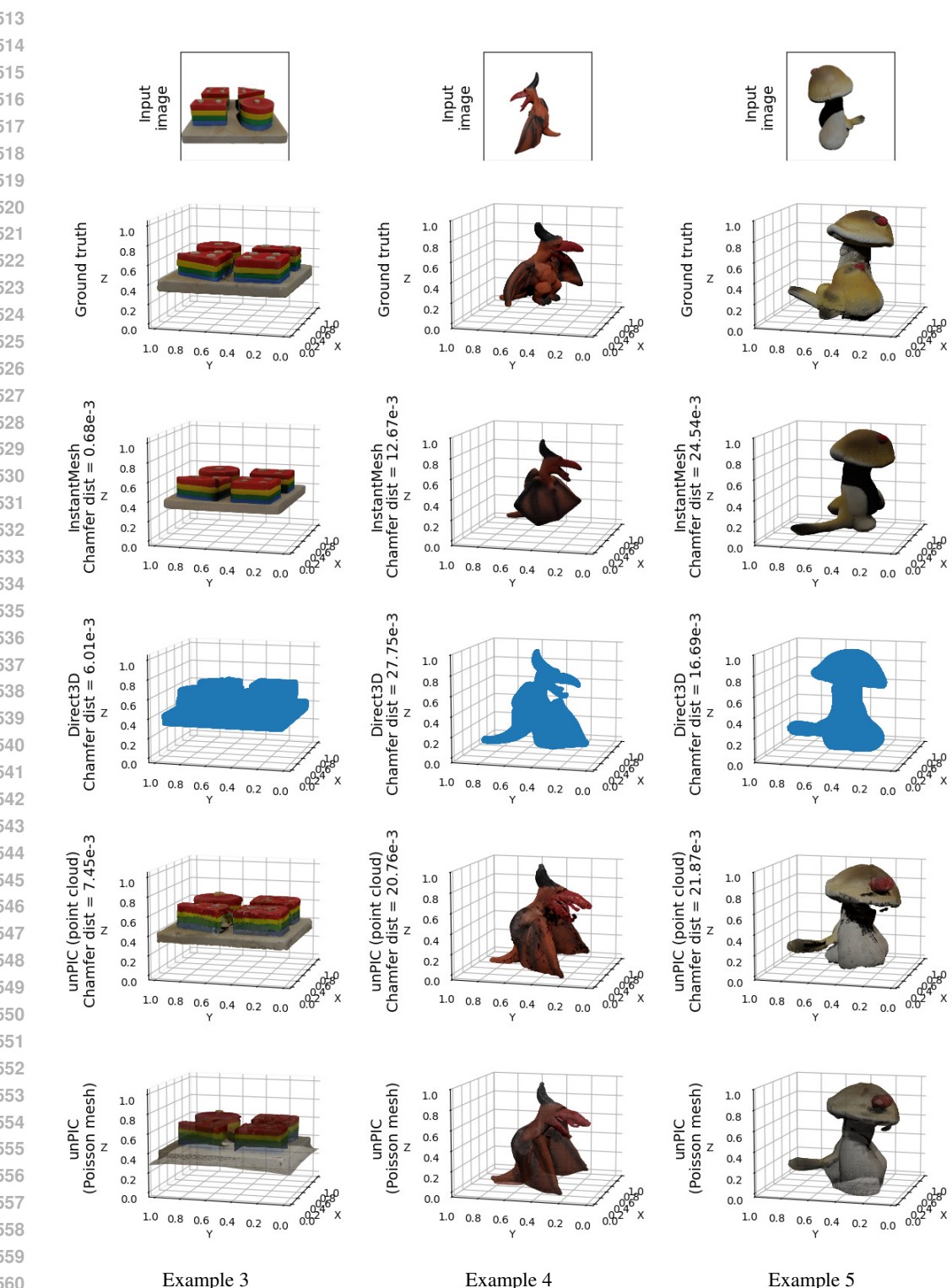

Figure 13: **3D reconstruction results from unPIC vs baseline methods (Part 2 of 14).** We show a 195° rotation of the object relative to the input image. The bottom row (unPIC's reconstructed mesh) is visualized with transparency to show the full 3D shape. (Figure continues on next page.)

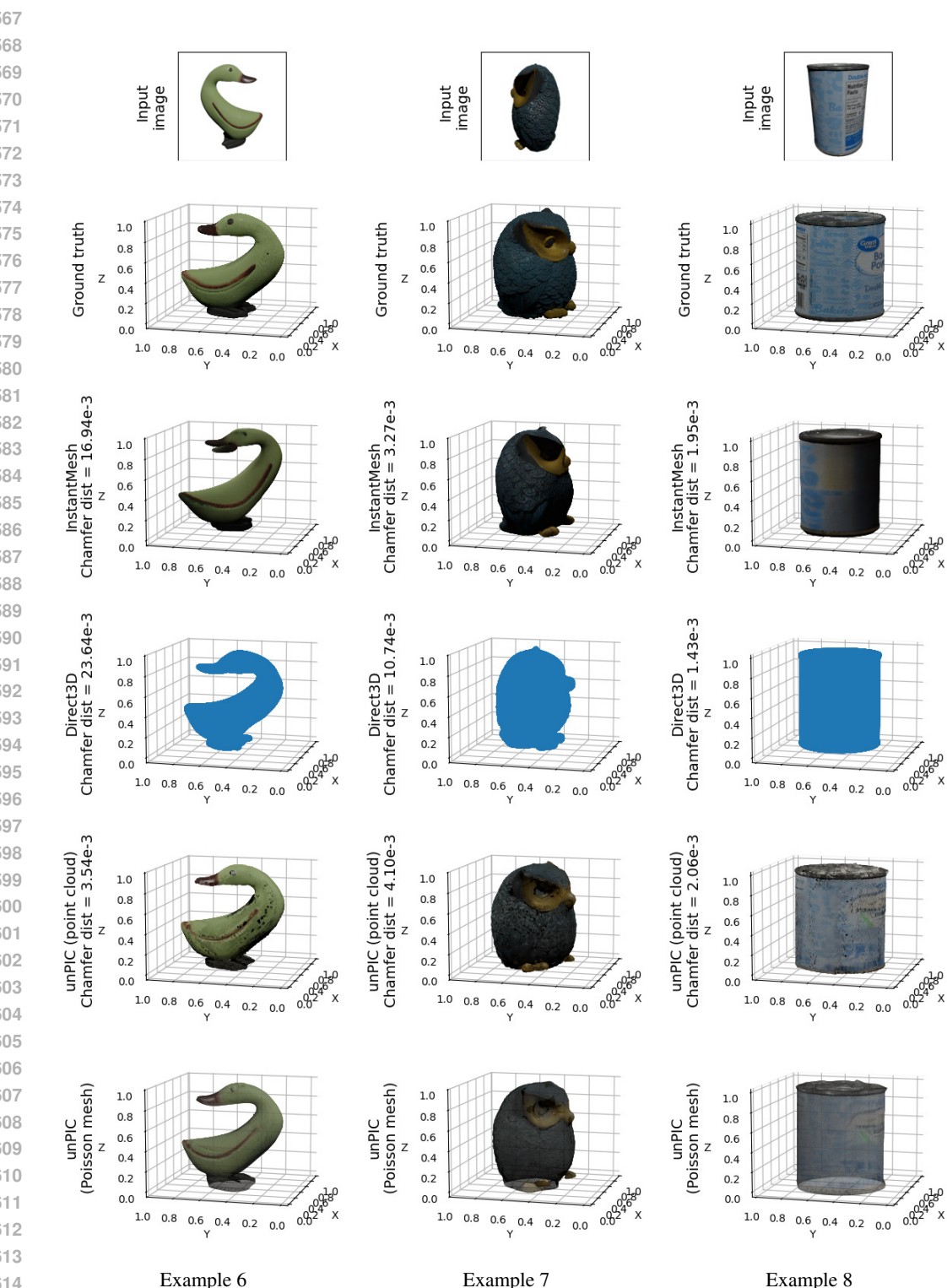

Example 6                    Example 7                    Example 8

Figure 13: **3D reconstruction results from unPIC vs baseline methods (Part 3 of 14).** We show a $195°$ rotation of the object relative to the input image. The bottom row (unPIC's reconstructed mesh) is visualized with transparency to show the full 3D shape. (Figure continues on next page.)

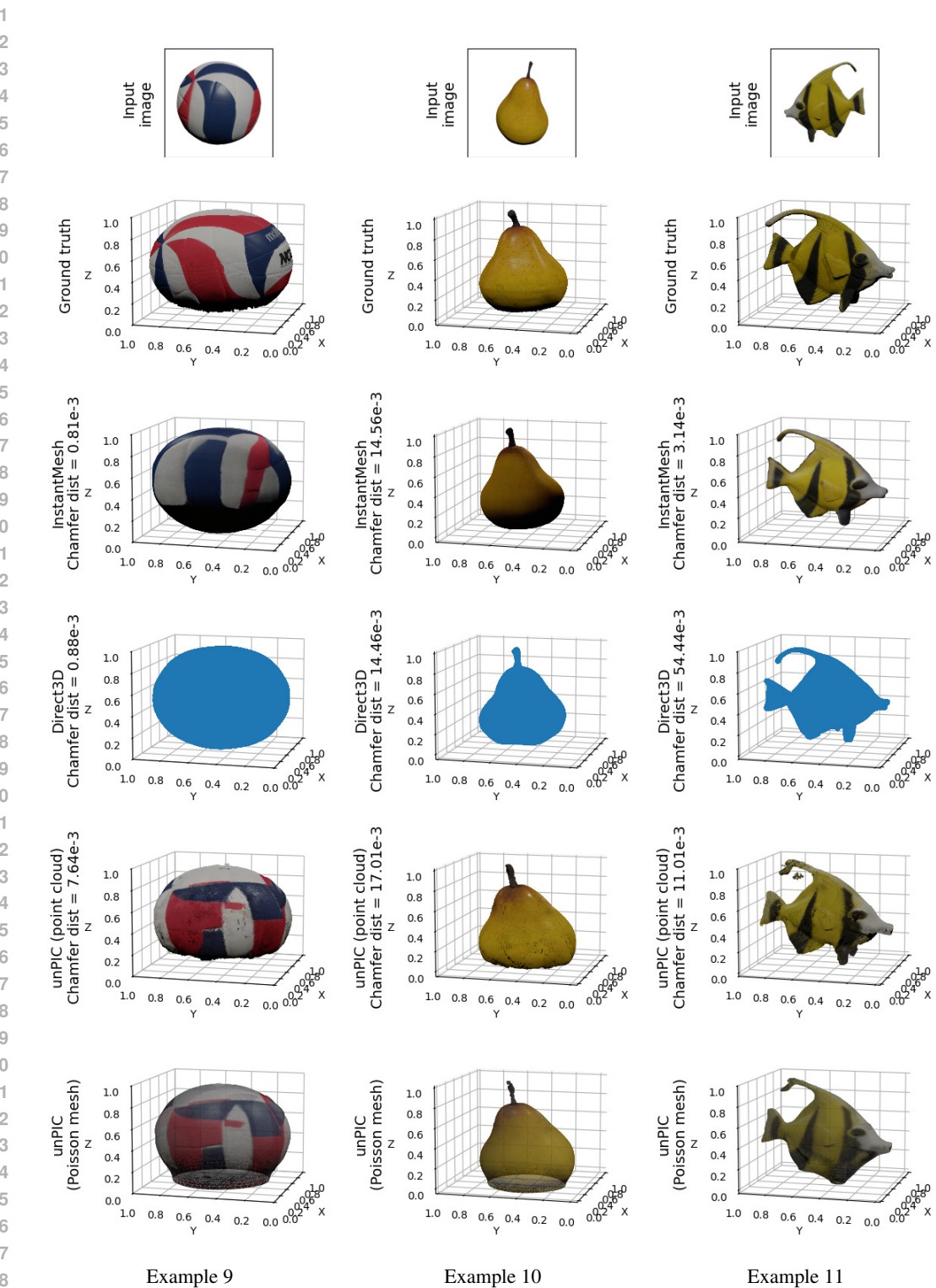

Figure 13: **3D reconstruction results from unPIC vs baseline methods (Part 4 of 14).** We show a 195° rotation of the object relative to the input image. The bottom row (unPIC's reconstructed mesh) is visualized with transparency to show the full 3D shape. (Figure continues on next page.)

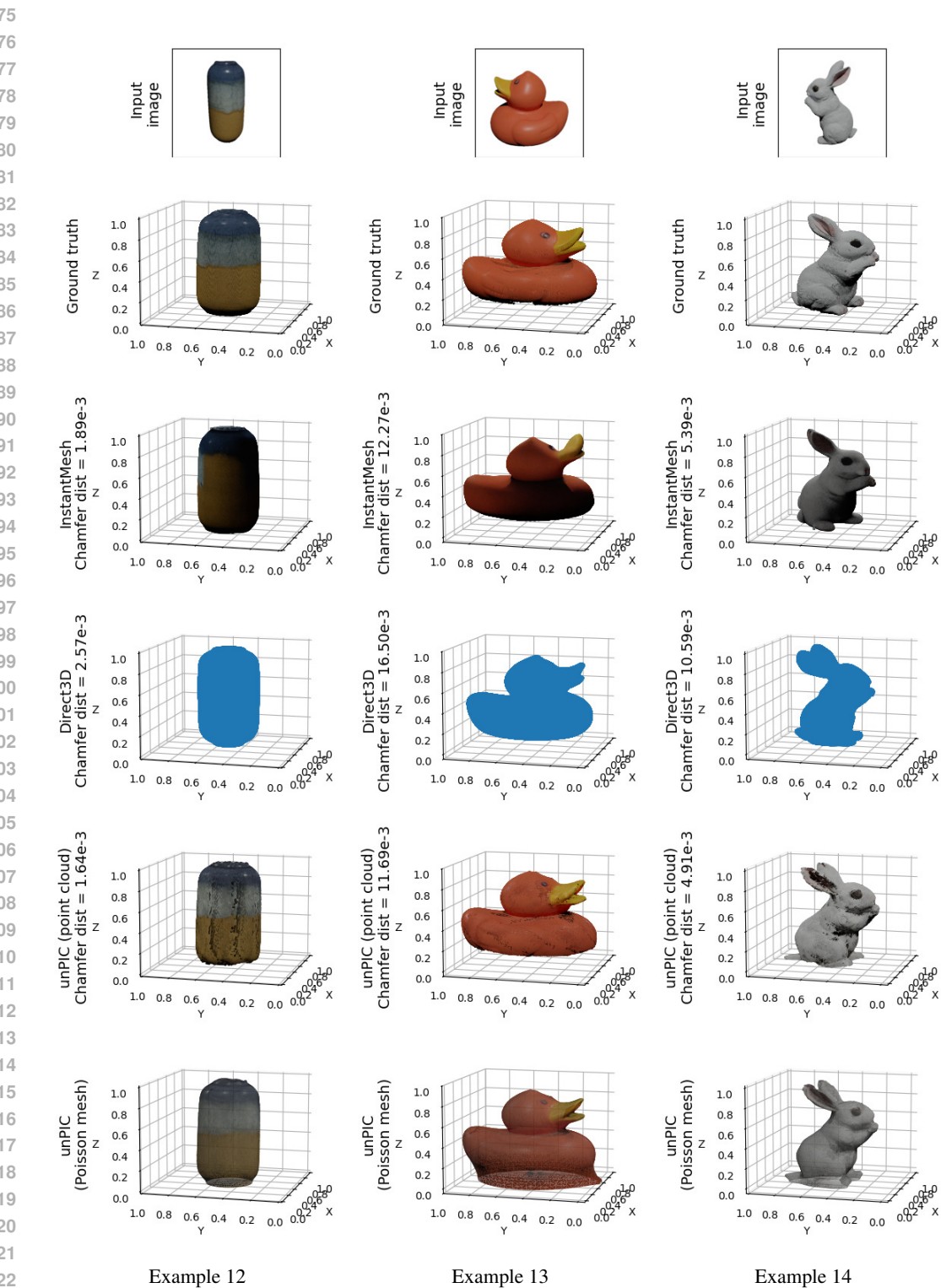

Example 12        Example 13        Example 14

Figure 13: **3D reconstruction results from unPIC vs baseline methods (Part 5 of 14).** We show a $195°$ rotation of the object relative to the input image. The bottom row (unPIC's reconstructed mesh) is visualized with transparency to show the full 3D shape. (Figure continues on next page.)

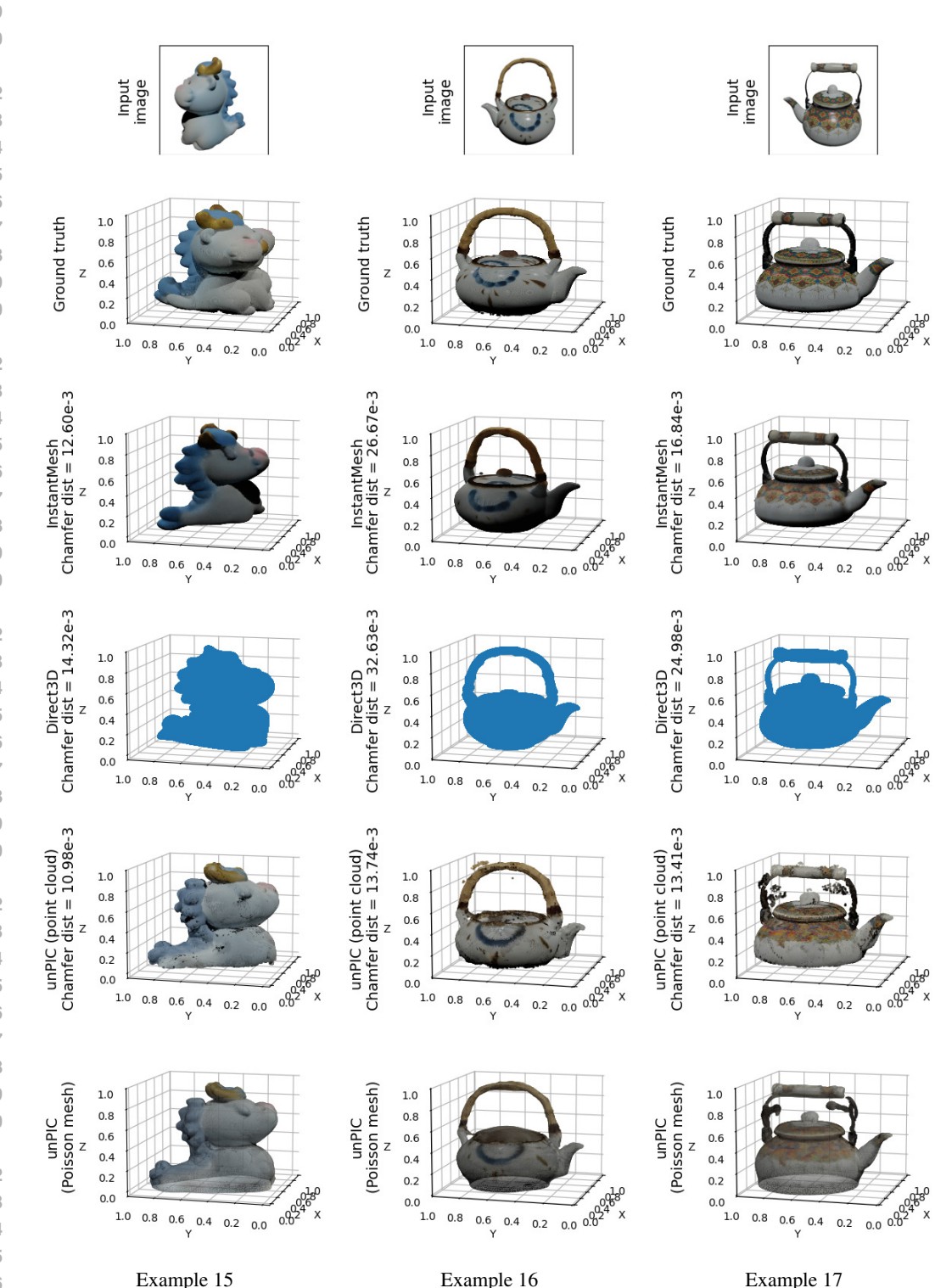

Example 15          Example 16          Example 17

Figure 13: **3D reconstruction results from unPIC vs baseline methods (Part 6 of 14).** We show a $195°$ rotation of the object relative to the input image. The bottom row (unPIC's reconstructed mesh) is visualized with transparency to show the full 3D shape. (Figure continues on next page.)

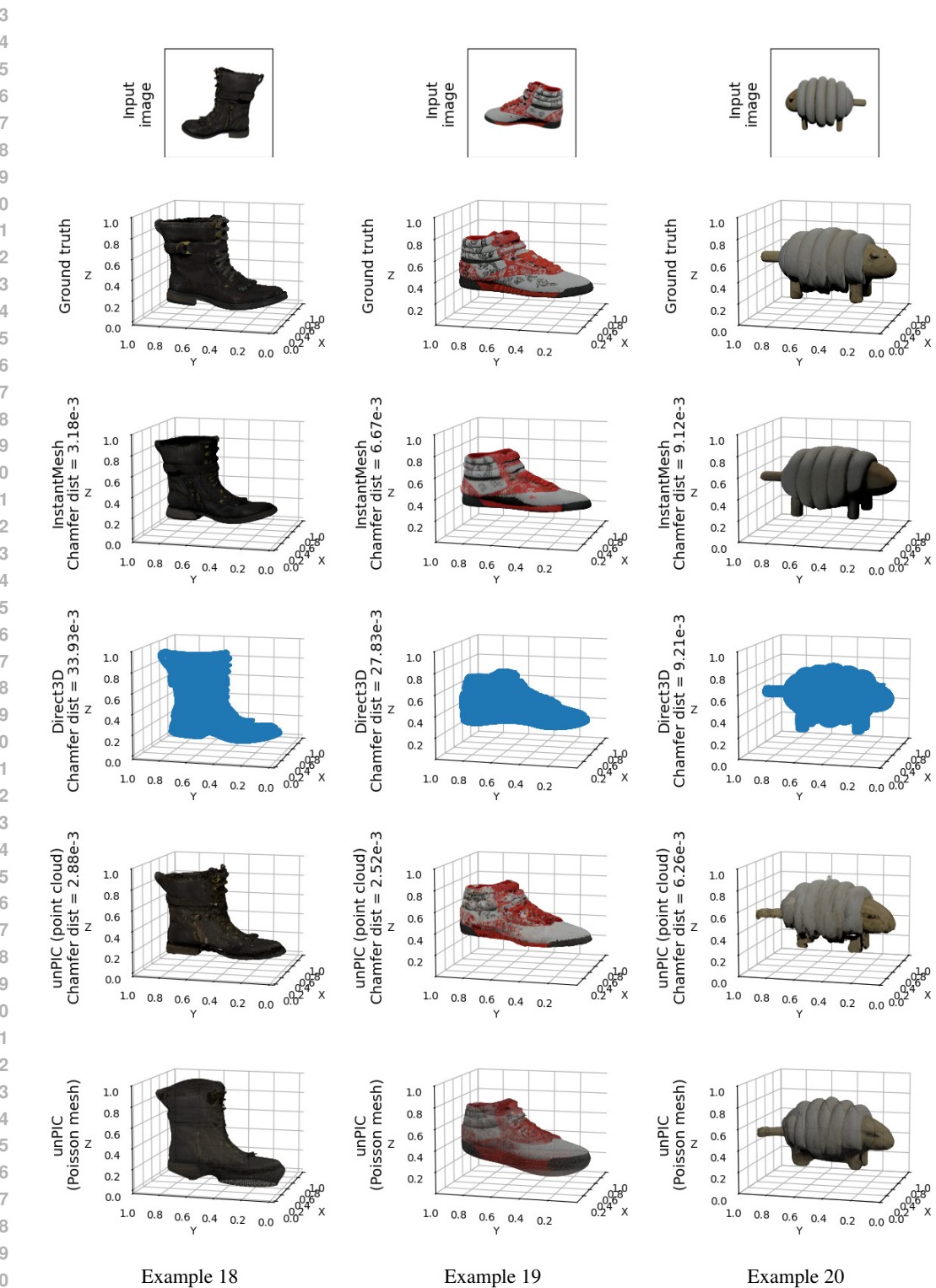

Example 18          Example 19          Example 20

Figure 13: **3D reconstruction results from unPIC vs baseline methods (Part 7 of 14).** We show a $195°$ rotation of the object relative to the input image. The bottom row (unPIC's reconstructed mesh) is visualized with transparency to show the full 3D shape. (Figure continues on next page.)

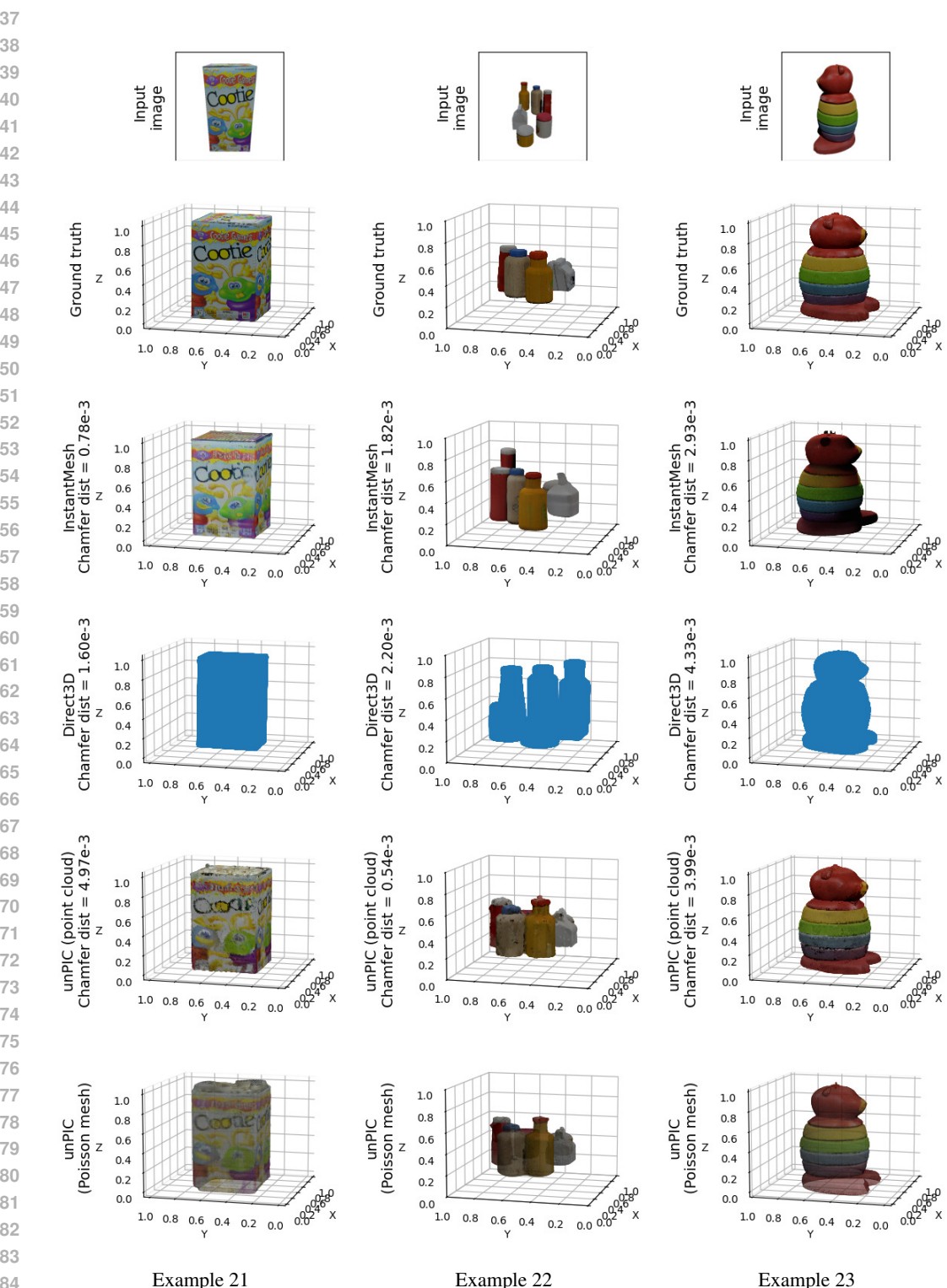

Example 21        Example 22        Example 23

Figure 13: **3D reconstruction results from unPIC vs baseline methods (Part 8 of 14).** We show a 195° rotation of the object relative to the input image. The bottom row (unPIC's reconstructed mesh) is visualized with transparency to show the full 3D shape. (Figure continues on next page.)

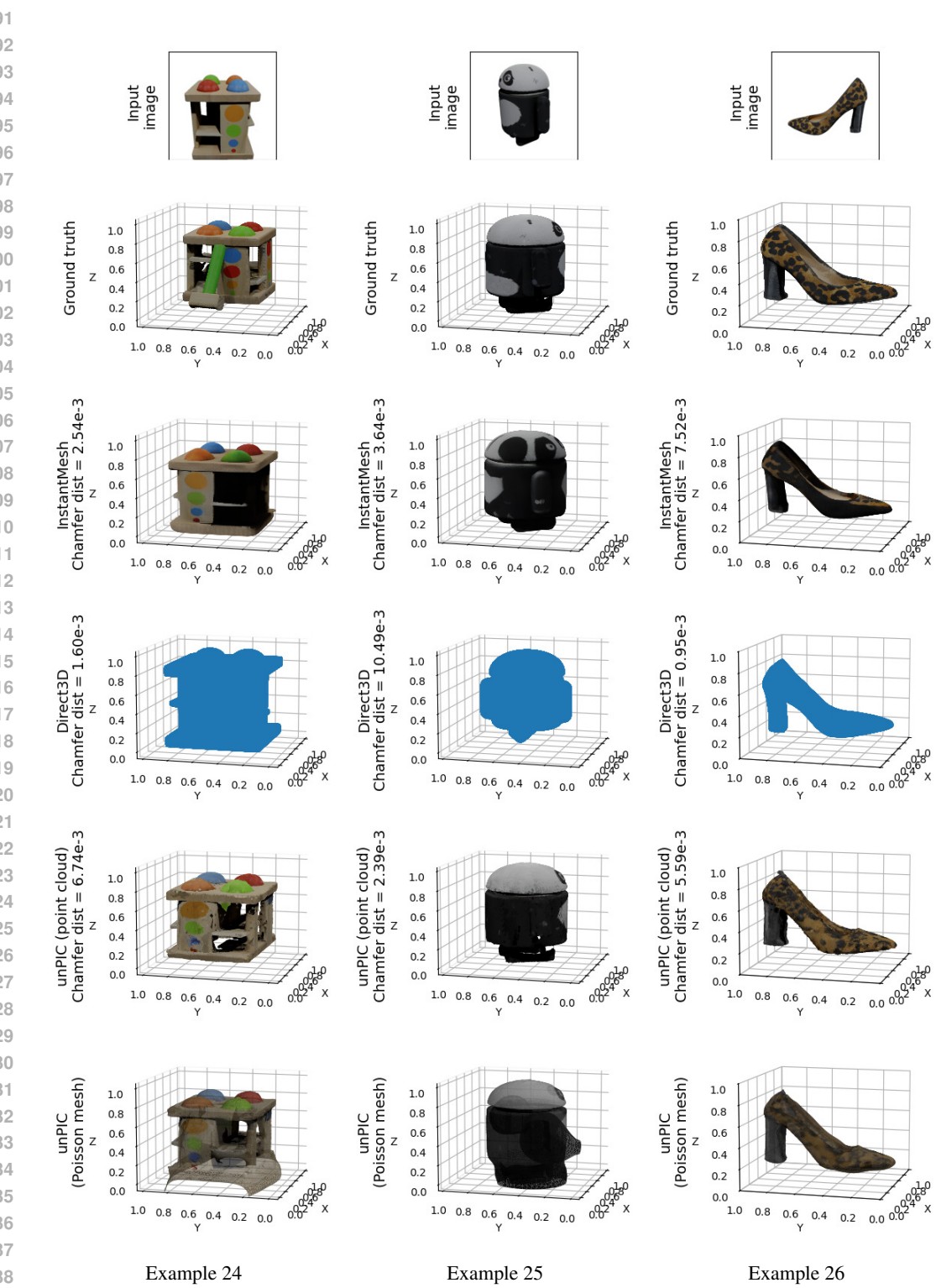

Example 24        Example 25        Example 26

Figure 13: **3D reconstruction results from unPIC vs baseline methods (Part 9 of 14).** We show a $195°$ rotation of the object relative to the input image. The bottom row (unPIC's reconstructed mesh) is visualized with transparency to show the full 3D shape. (Figure continues on next page.)

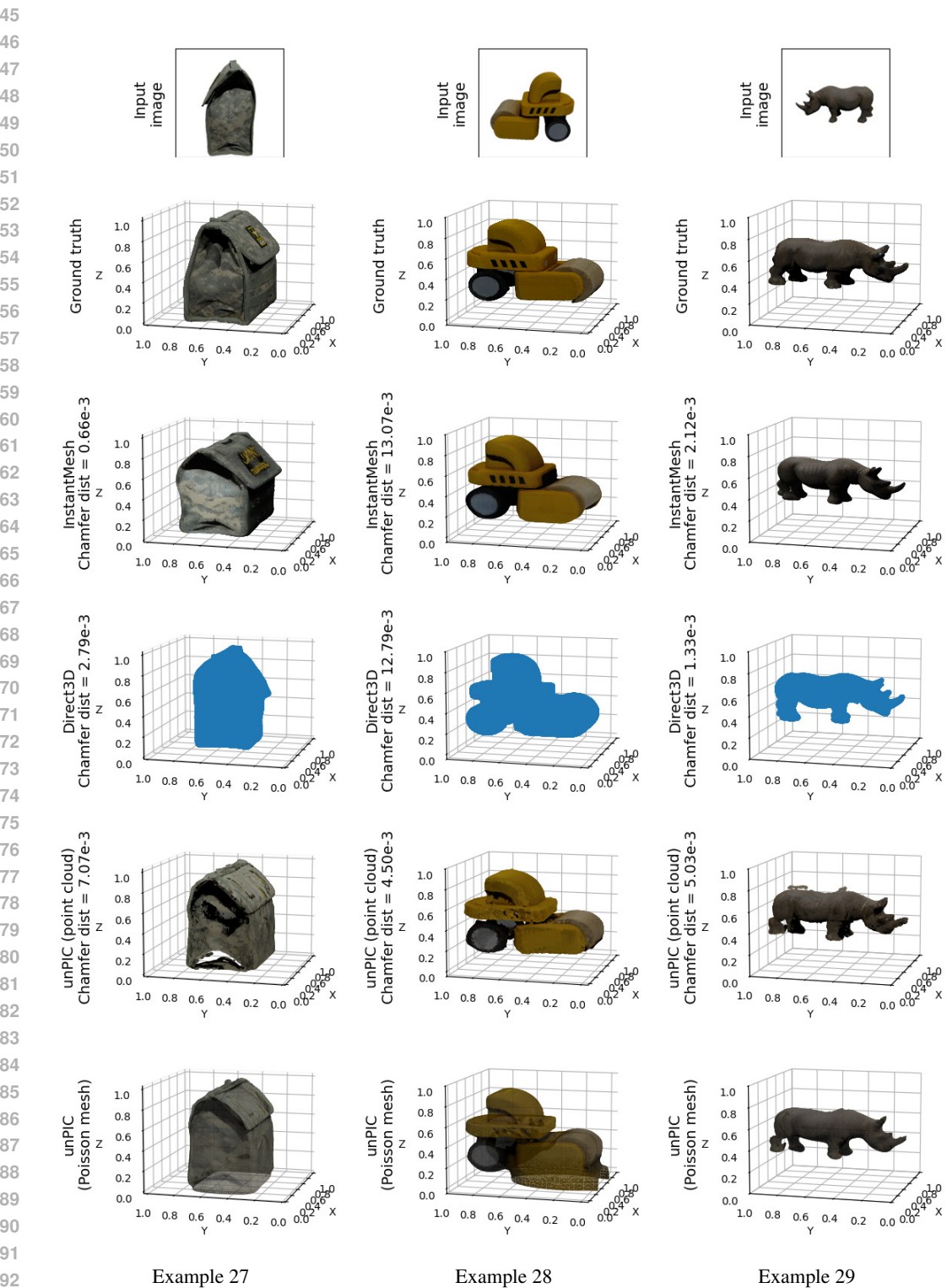

Example 27          Example 28          Example 29

Figure 13: **3D reconstruction results from unPIC vs baseline methods (Part 10 of 14).** We show a $195°$ rotation of the object relative to the input image. The bottom row (unPIC's reconstructed mesh) is visualized with transparency to show the full 3D shape. (Figure continues on next page.)

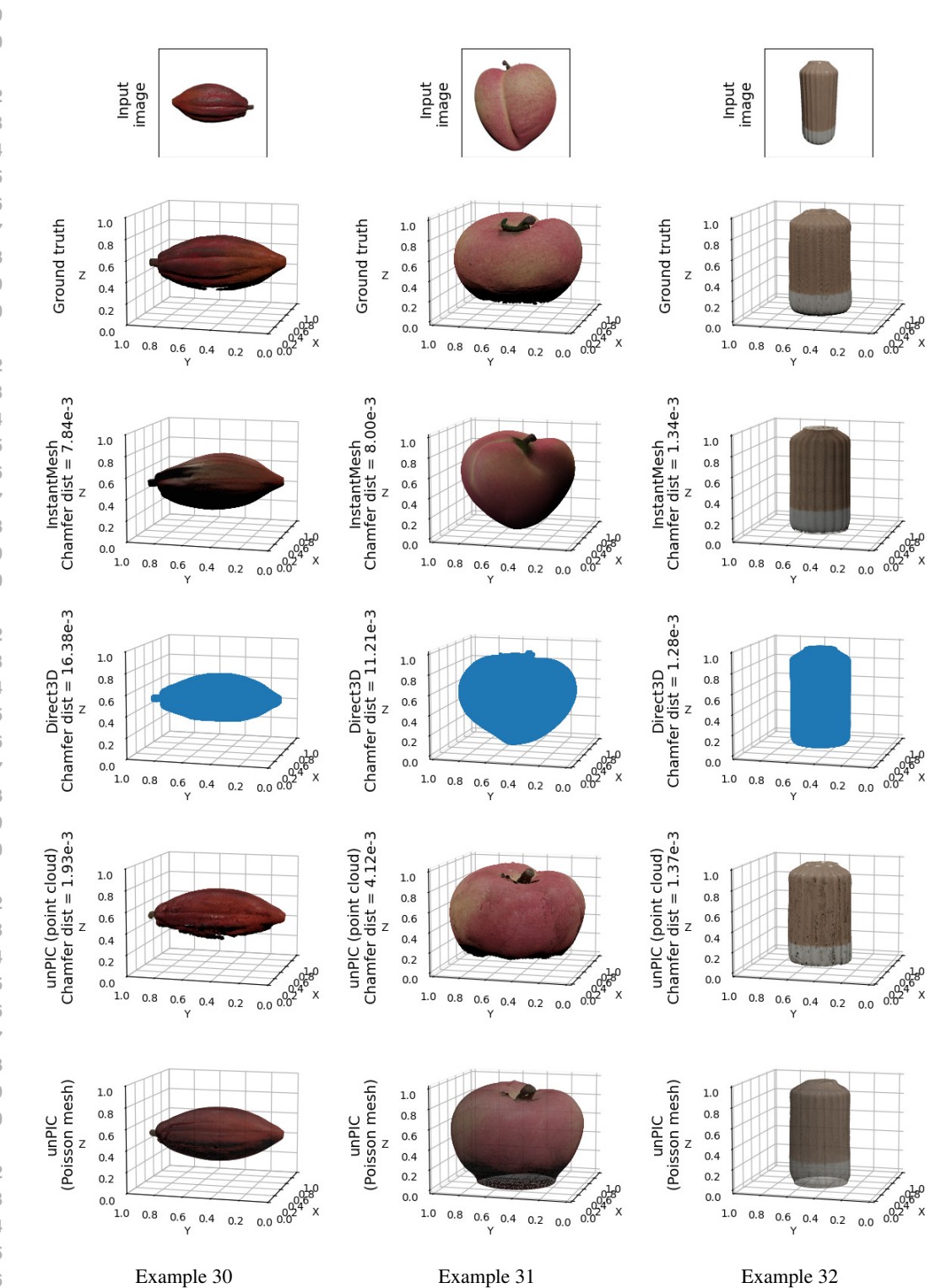

Example 30                     Example 31                     Example 32

Figure 13: **3D reconstruction results from unPIC vs baseline methods (Part 11 of 14).** We show a 195° rotation of the object relative to the input image. The bottom row (unPIC's reconstructed mesh) is visualized with transparency to show the full 3D shape. (Figure continues on next page.)

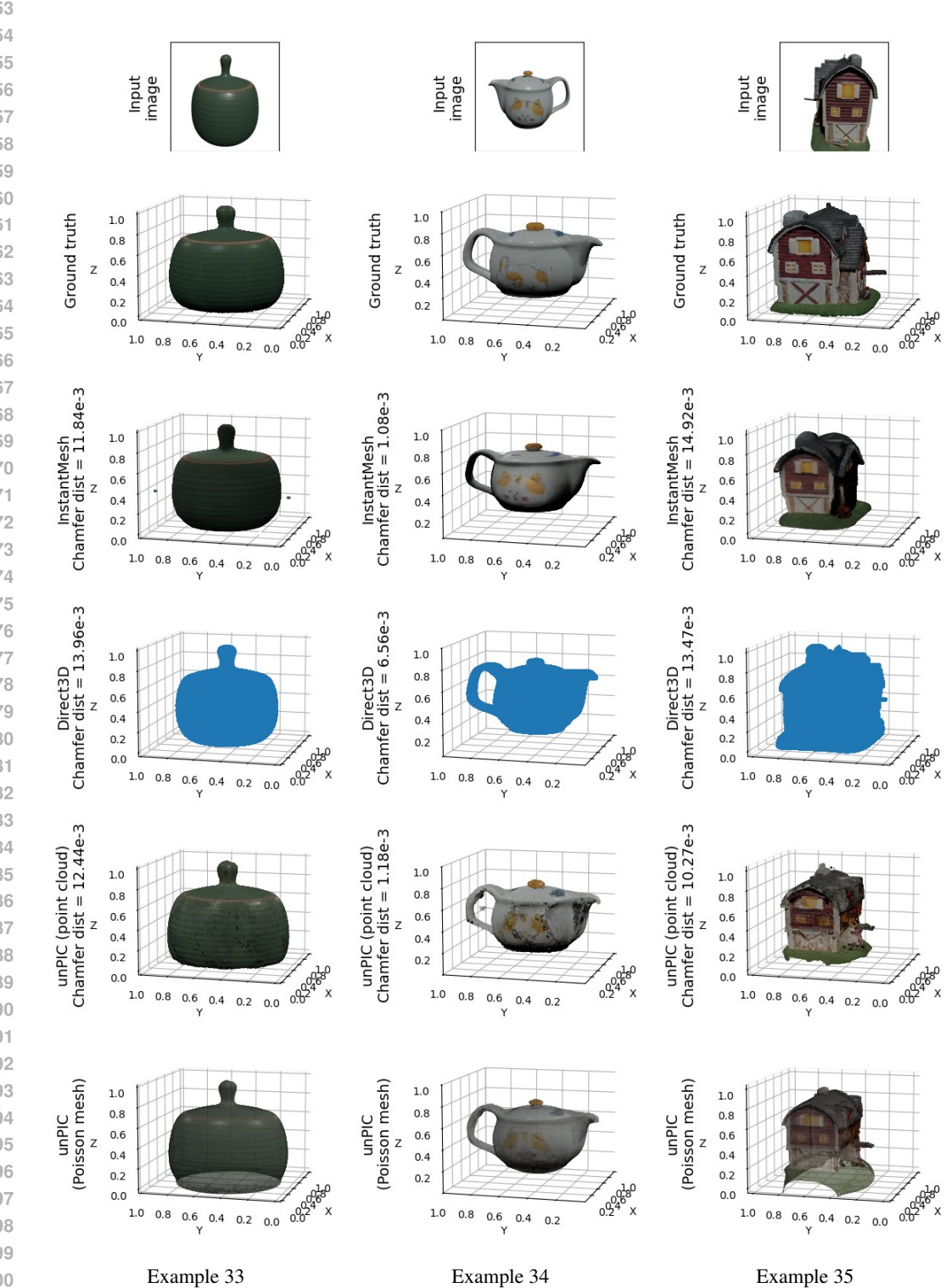

Example 33       Example 34       Example 35

Figure 13: **3D reconstruction results from unPIC vs baseline methods (Part 12 of 14).** We show a $195°$ rotation of the object relative to the input image. The bottom row (unPIC's reconstructed mesh) is visualized with transparency to show the full 3D shape. (Figure continues on next page.)

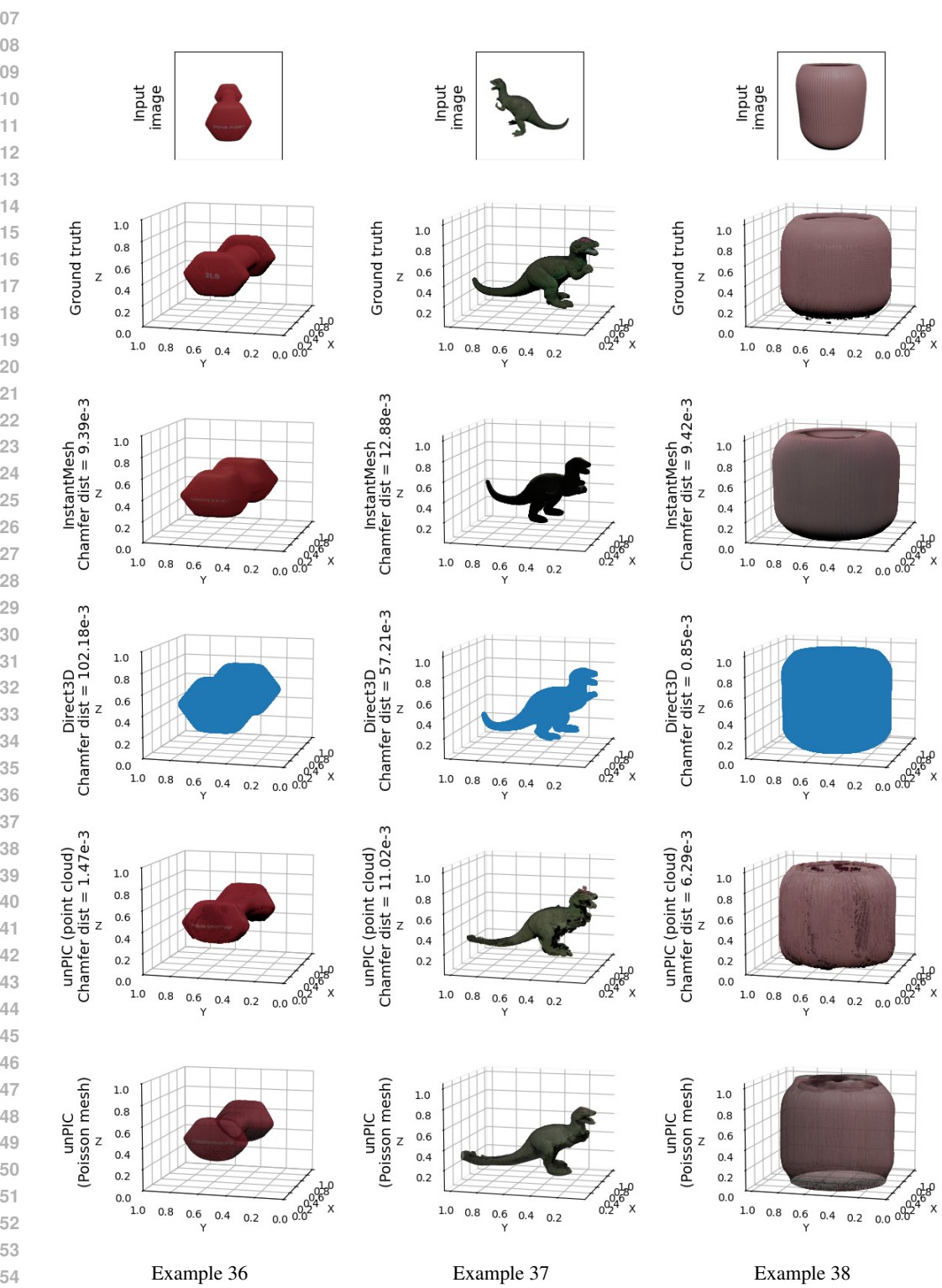

Example 36     Example 37     Example 38

Figure 13: **3D reconstruction results from unPIC vs baseline methods (Part 13 of 14).** We show a $195°$ rotation of the object relative to the input image. The bottom row (unPIC's reconstructed mesh) is visualized with transparency to show the full 3D shape. (Figure continues on next page.)

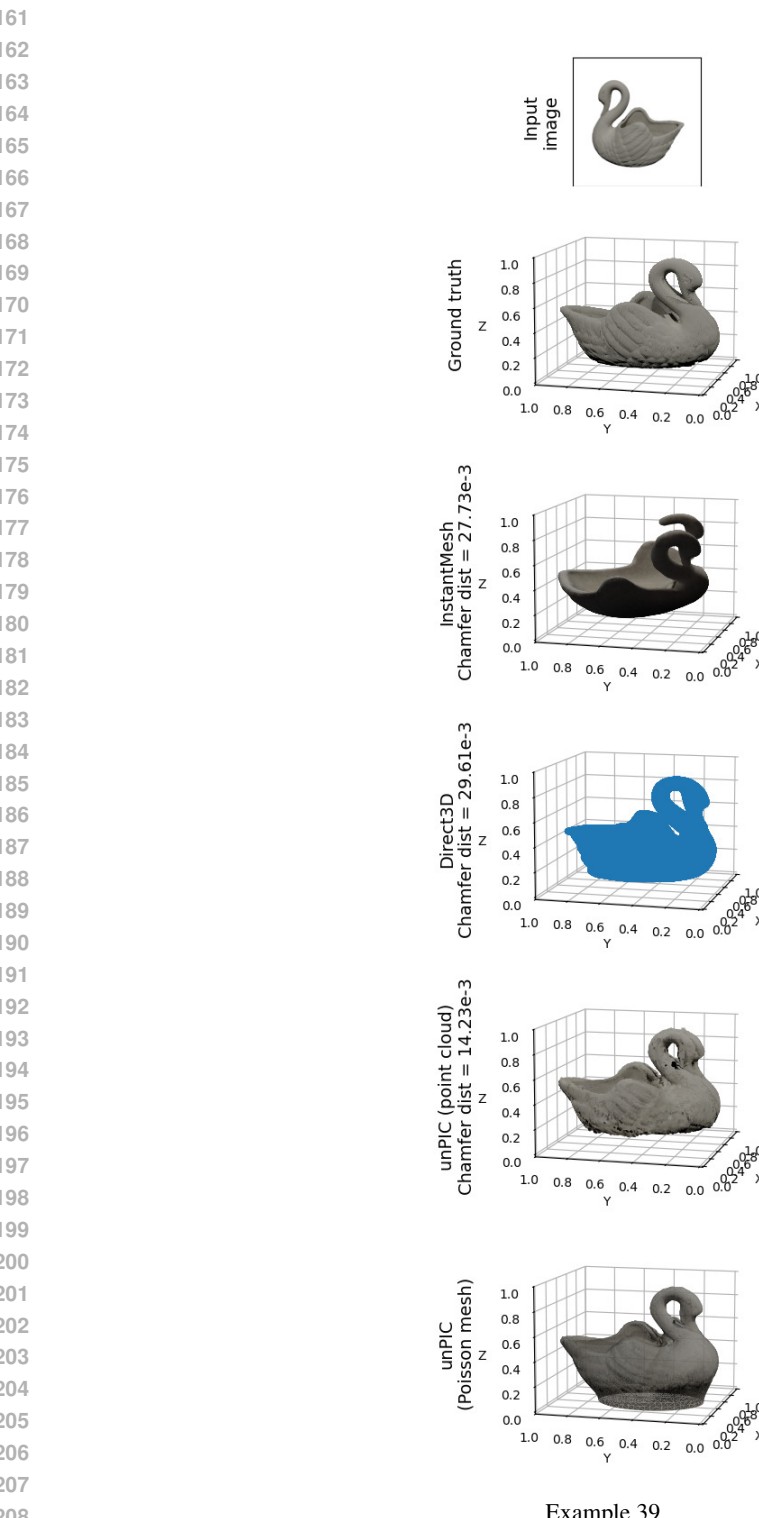

Example 39

Figure 13: **3D reconstruction results from unPIC vs baseline methods (Part 14 of 14).** We show a 195° rotation of the object relative to the input image. The bottom row (unPIC's reconstructed mesh) is visualized with transparency to show the full 3D shape.

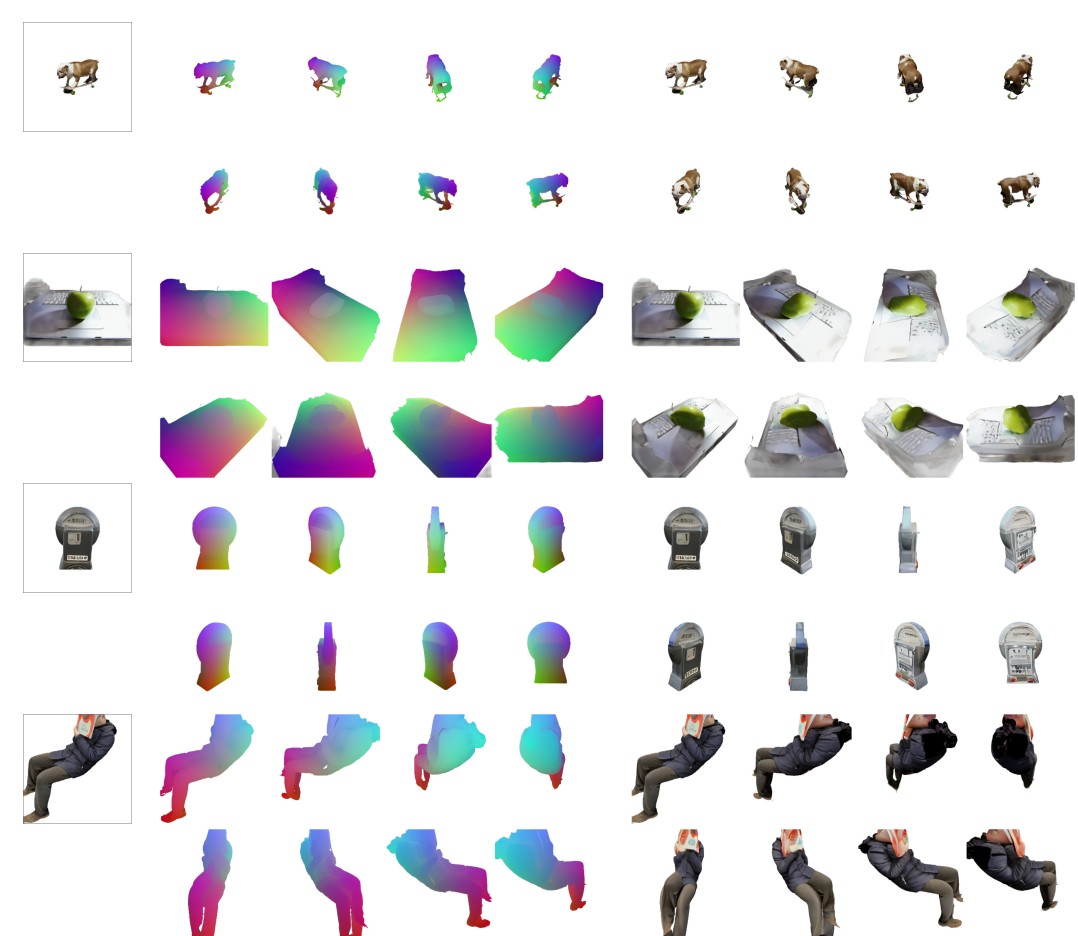

Figure 14: **unPIC on in-the-wild images.** Each pair of rows shows: the input image in the top-left box, predicted CROCS in the middle (2x4 grid), and predicted novel-view images to the right (2x4).

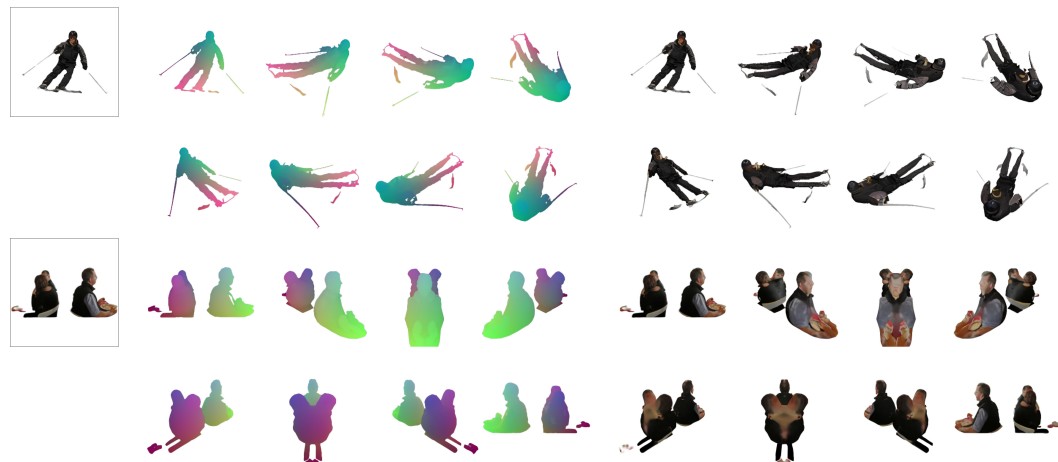

Figure 15: **Failure cases on in-the-wild images.** Each pair of rows shows: the input image in the top-left box, predicted CROCS in the middle (2x4 grid), and predicted novel-view images to the right (2x4). In the first example, unPIC rotates the person in the image plane, likely due to interpreting the source image as an overhead view. In the second example, unPIC predicts plausible poses, but fails to texture the images correctly, likely due to the out-of-distribution case of multiple humans in the scene.

