# OpenReview forum: "How to Spin an Object: First, Get the Shape Right"
_ICLR.cc/2026/Conference — Submitted to ICLR 2026_

### Official Review · Reviewer_Um9x · 2025-10-26

**Soundness:** 3
**Presentation:** 3
**Contribution:** 1
**Rating:** 2
**Confidence:** 5

**Summary:**

This paper proposes a new method to generate novel-view point maps and RGB images from single-view inputs. The key idea is to adopt multiview diffusion to generate both. Then, the point cloud could be directly extracted from the generated point maps and RGB images. Finally, the performance is evaluated on the novel view synthesis and 3D reconstruction tasks, outperforming baselines like Free3D, One-2-3-45 and OpenLRM.

**Strengths:**

The whole pipeline is reasonable and authors successfully train the model and demonstrate the performance.

**Weaknesses:**

1. The idea of generating point maps has already been explored by SweetDreamer (ICLR'23) two years ago. Some recent works, like World-consistent Video Diffusion with Explicit 3D Modeling (CVPR'25), also use this idea. The paper does not discuss the difference. Some very similar papers about multiview diffusion papers, like MVDream, SyncDreamer, and Wonder3D, are not included in the discussion either. The method proposed by the paper is already well-studied in these existing works.
2. Another main problem is that the paper seems to miss a whole set of papers about latent vecset diffusion, like CLAY, Hunyuan, TripoSG, and so on, which could produce much better results than the proposed method.

In summary, the idea is already well-explored by existing works, and the authors are encouraged to read these papers and include more discussion on the differences from the existing works.

**Questions:**

N/A

---

> ### Author Response · Authors · 2025-11-18
> **Initial Response (1/2)**
>
> We sincerely thank the reviewer for their time and for highlighting prior works like SweetDreamer, MVDream, and SyncDreamer. We respectfully disagree that our method is "already well-explored" by these works. While they share the goal of 3D generation, **unPIC introduces a fundamentally different hierarchical paradigm** (geometry-first generation vs. joint generation or optimization).
>
> Below, we differentiate our contributions and demonstrate that we have, in fact, empirically evaluated the core ideas of SweetDreamer within our paper and found our proposed method (CROCS) to be superior.
>
> # 1. Differentiation from SweetDreamer (ICLR '23)
> The reviewer states that the idea of generating point maps was explored by SweetDreamer. We carefully analyzed SweetDreamer and identified three critical differences that make our CROCS representation distinct and more suitable for feed-forward generation:
> ### A. Reference Frame (Object-Centric vs. Camera-Relative):
> - **SweetDreamer (CCM):** Relies on "Canonical Coordinate Maps" (CCM) which assume **dataset-wide category alignment.** As stated in their paper: *"we assume that all objects within the same category adhere to a canonical orientation in the training data".* Their coordinate system is fixed to the object (e.g., the front of a car is always +Y).
> - **unPIC (CROCS):** Uses Camera-Relative coordinates. The coordinate system is anchored to the *source camera*. This acts as a geometric data augmentation, which in contrast to SweetDreamer, makes our model **pose-agnostic** and capable of handling unaligned datasets or multi-object scenes where no single "canonical" front exists.
>
> ### B. Scaling (Anisotropic vs. Isotropic):
> - **SweetDreamer:** Uses anisotropic scaling, normalizing $x, y, z$ independently to $[0,1]$. This distorts the object's metric shape (e.g., a flat plate becomes a cube in feature space) to maximize gradients for optimization.
> - **unPIC:** Uses isotropic (uniform) scaling. We preserve the object's true metric aspect ratio. This is critical for our generative task: it ensures the predicted "blueprint" is geometrically faithful, enabling the **direct point-cloud extraction** shown in Figure 2 without requiring regressed bounding-box parameters to "un-distort" the geometry.
>
> ### C. Empirical Superiority (The "NOCS" Baseline):
> We actually evaluated the SweetDreamer-style representation in our paper under the name "NOCS" (Sec 4.1), which we defined as using the "default creator-intended pose" (object-centric alignment).
> - **Table 1b:** We show that predicting object-centric maps (SweetDreamer/NOCS) from a single image is up to **10x harder** (MSE 11.94 vs 1.21) than predicting our camera-relative CROCS. The model struggles to hallucinate the arbitrary "canonical front" of the object.
> - **Table 1a:** Using object-centric maps yields worse novel-view synthesis quality (MSE 5.318) compared to CROCS (MSE 4.914).
>
> Thus, our results in Table 1 demonstrate that our camera-relative approach (CROCS) outperforms the object-centric approach used in SweetDreamer.
>
> &#9733; **Action Item:** Given this functional equivalence, **would you recommend we rename the 'NOCS' baseline in Table 1 and Section 4.1 to 'Object-Centric, Isotropic CCM (SweetDreamer)'?** We are happy to make this change if it makes the comparison to SweetDreamer clearer for future readers.
>
> # 2. Optimization vs. Feed-Forward Generation
> SweetDreamer uses the point-based maps as a supervision signal (SDS loss) to optimize a separate 3D representation (NeRF/DMTet). It does not generate 3D directly.
>
> In contrast, unPIC generates the CROCS images as an explicit intermediate blueprint. This allows us to extract a point cloud directly from the inference output (i.e., without further processing), bypassing the slow per-scene optimization required by SweetDreamer.
>
> To summarize, **while 3D generation is optimization-based in SweetDreamer, it is feed-forward in unPIC.**
>
> # 3. Comparison with MVDream, SyncDreamer, and Wonder3D
>
> The reviewer notes similarity to multiview diffusion models.
> - **Joint vs. Disentangled:** MVDream and SyncDreamer generate appearance and (implicit) geometry jointly. unPIC is hierarchical: we disentangle $p(geometry)$ and $p(appearance | geometry)$.
> - **Consistency:** By generating geometry first, unPIC creates a "blueprint" that enforces consistency. In **Table 3**, unPIC outperforms EscherNet and Free3D (which are multiview diffusion models similar to the ones mentioned) on IoU and consistency metrics.
> - **Direct 3D output:** Unlike MVDream (which produces images requiring subsequent reconstruction), unPIC produces a point cloud natively.

---

> ### Author Response · Authors · 2025-11-18
> **Initial Response (2/2)**
>
> # 4. Latent Vecset Diffusion (CLAY, TripoSG, Hunyuan)
>
> We acknowledge these recent works. Vecsets are indeed promising--our related work included CraftsMan3D, which builds on 3DShape2VecSet.
>
> We present a brief table comparing CLAY, TripoSG, Hunyuan3D-2, and another method TRELLIS with our work below (if it helps, we will expand this table and add it to our paper):
>
> ```
> +--------------+---------------------------------------------+-----+------------------------------------------+
> |    Model     |           Core 3D Representation            | VAE |              Key Advantage               |
> +--------------+---------------------------------------------+-----+------------------------------------------+
> | CLAY         | Featured Point Cloud (Multi-resolution)     | 3D  | Topological freedom; Multi-modal control |
> | TripoSG      | Implicit SDF (Signed Distance Function)     | 3D  | Inference Speed; Watertight Manifold     |
> | Hunyuan3D-2  | Vector Sets (Sequence of latent vectors)    | 3D  | Texture fidelity (4K); Decoupled control |
> | TRELLIS      | Structured LATents (Sparse voxels + feats.) | 3D  | Output versatility (Mesh, 3DGS or NeRF)  |
> | unPIC (ours) | CROCS (Camera-Relative Object Coordinates)  | 2D  | Direct 3D via View Synthesis             |
> +--------------+---------------------------------------------+-----+------------------------------------------+
> ```
>
> One insight from this comparison is that although the field is converging on Generative Architectures (Transformers + Flow/Diffusion), it remains deeply divided on 3D Representations:
> 1. **Continuity vs. Discreteness:** TripoSG and Hunyuan3D-2 (via SDF and Vector Sets) favor continuous, mathematically smooth representations. This suits objects with defined surfaces (cars, furniture). CLAY and TRELLIS (via Points and Sparse Grids) embrace discreteness, allowing for more chaotic, high-frequency, or disjointed structures (vegetation, fur). unPIC falls in the latter category.
> 2. **The VAE Dimension:** While the other four models invest heavily in 3D-native VAEs (learning to compress voxels or points directly), unPIC argues that CROCS allows standard 2D VAEs (like Stable Diffusion 1.4, fine-tuned on tiled images) to handle 3D geometry effectively. This avoids the complexity of training 3D-native backbones; we use a standard 2D UNet and rely on the coherence of multiview diffusion.
> 3. **The Texture Bottleneck:** Hunyuan3D-2’s texture quality reinforces our point that decoupling is superior to end-to-end learning for appearance. Learning the joint distribution of geometry and texture simultaneously (as TripoSG attempts) forces the model to compromise. By treating texture as a "painting" problem on top of a fixed mesh, Hunyuan3D-2 leverages the mature power of 2D image generators. unPIC leverages the same insight.
>
> We further note that Hunyuan3D (versions 1.0, 2.0, and 2.5) and TripoSG remain *unpublished*--they are only available on Arxiv.
>
> # Conclusion
> We believe unPIC offers a distinct contribution: a **geometry-first, camera-relative** generative prior that outperforms the object-centric optimization maps of SweetDreamer, and the joint-generation paradigms of current multiview diffusion models. unPIC introduces a notable counter-point by utilizing a 2D VAE. We hope this clarification on the fundamental differences—supported by our ablation studies in Table 1—addresses your concerns regarding novelty.
>
> We’d be grateful to know if there is anything else we can address. Thank you very much.

---

> > ### Comment · Reviewer_Um9x · 2025-11-18
> >
> > Thanks for your reply!
> >
> > However, the relative coordinate map generation/prediction is also explored by https://github.com/EnVision-Research/LucidFusion?tab=readme-ov-file
> > And there are also some related works like https://ml.cs.tsinghua.edu.cn/~zhengyi/CRM/, also using coordinate map generation.
> >
> > This makes me feel that the related work review of this paper is not good enough.
> > Given all these works that generate coordinate maps, the idea in this paper to first generate a relative coordinate map seems to be incremental and a relatively minor improvement.

---

> ### Author Response · Authors · 2025-11-19
> **Further thoughts on novelty**
>
> Many thanks for the additional references. We will include LucidFusion and CRM in our related work and connect them to other methods that rely on coordinate maps. After analyzing the papers, we still see a substantial difference in both the problem setting and methodology, most critically the fact that _neither of them follows a geometry-first approach_:
>
> - **CRM** employs a `"single image → multiview images → multiview pointmaps → 3D mesh"` pipeline. In contrast, unPIC’s generative process (`"single image → multiview pointmaps → multiview images → direct 3D point cloud"`) is closer in spirit to the **geometry-first approach** of Hunyuan3D-2, CLAY, and TRELLIS. The choice of pointmaps is also different: CRM uses canonical (object-centric) coordinate maps, forcing it to hallucinate an object-specific canonical orientation that is not tied to the input camera. In Table 1, we showed that a “NOCS / object-centric CCM” baseline is inferior to "CROCS," our camera-relative approach, for single-view inference.
>
>   CRM’s main novelty lies in the mesh-prediction part of its pipeline. unPIC's key advantage, on the other hand, is the ability to obtain a reliable 3D point cloud ***directly*** from the generated multiview images (CROCS + RGB). Our point-cloud geometric accuracy (enabled by CROCS, see Table 5) justifies a simpler approach to mesh reconstruction (without additional neural networks, e.g., using Screened Poisson or Ball Pivoting) than CRM.
>
> - **LucidFusion:** We note that this is paper is very recent: it was published at CGF on 11 October 2025, i.e., after our ICLR submission in September 2025. While this work indeed uses relative coordinates rather than canonical object-centric coordinates, it focuses on 3D Gaussian Splatting to handle unposed multi-view alignment. In their single-image-to-3D setting, novel views are first generated using an external multiview diffusion model (Flux), and LucidFusion then predicts pointmaps and a 3DGS representation from this multiview set. In contrast, our approach is itself a single-image → multiview diffusion model that factorizes generation into geometry (CROCS) and then appearance, and produces a 3D point cloud directly as part of this multiview synthesis process, without a separate reconstruction backbone.
>
> To our knowledge, there are very few papers that generate an accurate **3D point cloud from a single image directly as part of multiview synthesis**. Two older papers [1, 2] predate diffusion models, and were based on depth maps rather than coordinate maps. We believe unPIC is the first in effectively utilizing camera-relative coordinates for explicit point cloud generation in a feed-forward manner.
>
> [1] Wiles, Olivia, et al. "SynSin: End-to-end View Synthesis from a Single Image." Proceedings of the IEEE/CVF conference on computer vision and pattern recognition. 2020.
>
> [2]  Le, Hoang-An, et al. ‘Novel View Synthesis from Single Images via Point Cloud Transformation’. Proceedings of the British Machine Vision Conference (BMVC), 2020.
>
> Once again, we thank you for the opportunity to clarify our approach.

---

> > ### Comment · Reviewer_Um9x · 2025-11-20
> >
> > Thanks for your reply. I could improve my score to borderline reject. I understand the difference, but the contribution is a little bit incremental for me.

---

> > > ### Author Response · Authors · 2025-12-03
> > > **Thank you**
> > >
> > > Thank you for your early engagement and for updating your score to a 4. We appreciate the opportunity to clarify the positioning of our work. We have uploaded a revised pdf with an expanded Related Work section; it directly addresses the distinction between unPIC and all the prior publications you mentioned (SweetDreamer, CRM, CLAY, TRELLIS).

---

### Official Review · Reviewer_BpYY · 2025-11-01

**Soundness:** 3
**Presentation:** 3
**Contribution:** 3
**Rating:** 8
**Confidence:** 3

**Summary:**

The paper introduces unPIC, a method for generating a fully 3D-consistent spin of an object from a single input image by explicitly separating the prediction of underlying 3D geometry from textured appearance.

This hierarchical generation is implemented using two independently trained diffusion models: a multiview geometry prior, followed by a multiview appearance decoder. A key contribution enabling this architecture is a novel geometric representation called CROCS (Camera-Relative Object Coordinates), which provides dense pointmaps encoding per-pixel 3D coordinates anchored to the source camera.

The predicted geometry serves as a blueprint to coordinate the final views, enforcing consistency and enabling the direct generation of a 3D point cloud without a separate post-hoc reconstruction step.

This geometry-driven framework significantly outperforms leading methods on novel-view quality, geometric accuracy, and multiview consistency.

**Strengths:**

- The paper introduces CROCS (Camera-Relative Object Coordinates), a novel dense pointmap representation that is critical to the method's success. Empirical evidence strongly supports its effectiveness.
  - By conditioning the appearance decoder on CROCS, the framework allows for direct 3D generation. The output multiview CROCS images provide the vertices, and the RGB images provide the vertex colors, which assemble directly into a colored point cloud, bypassing the need for a separate post-hoc reconstruction step common in other pipelines.
- unPIC demonstrates superior quality, consistently outperforming strong contemporary baselines. The authors provide extensive experiments that substantiate the method’s performance advantages, and the model exhibits robust generalization to challenging real-world captures.
- The paper is easy to follow, with clear writing.

**Weaknesses:**

- The authors made a deliberate design choice not to canonicalize for changes in camera elevation. While this choice aligns with typical human mental rotation habits, it forces the model to implicitly infer the camera elevation from the source image. This implicit reliance on the appearance module to deduce a crucial geometric parameter is a source of fragility, which is confirmed by the observed failure case where the model misinterprets the source image (e.g., as an overhead view) and performs incorrect planar rotation.

- Both the geometry prior and the appearance decoder are implemented as multi-view diffusion models (MVD) and trained separately. This hierarchical two-stage MVD architecture, totaling 1.1 Billion parameters, is inherently computationally expensive.

**Questions:**

- Why did the CROCS VAE have a significantly lower KL divergence before fine-tuning compared to the RGB VAE (20823 vs 28005)? Does this suggest that the latent space of the CROCS representation is inherently smoother or closer to a standard Gaussian, contributing directly to CROCS's superior predictability?
- Could the authors provide a detailed breakdown of the total 1m13s inference time? Specifically, what proportion is spent in the Geometry Prior module versus the Appearance Decoder module? Such a decomposition would be valuable for guiding subsequent runtime optimizations.
- unPIC provides "Direct 3D" output by combining CROCS vertices and RGB colors into a colored point cloud. Unlike geometry-supervised pipelines that return explicit surfaces, the point cloud is not a ready-to-use explicit surface, potentially necessitating further reconstruction steps for applications requiring watertight meshes. Given that the output is a high-accuracy colored point cloud, have the authors explored post-processing to convert this point cloud into an explicit mesh? If so, what are the resulting mesh quality and the practical utility for downstream applications?

- A key advantage claimed for unPIC is its hierarchical approach designed to maximize diversity by sampling multiple geometries and appearances; however, the main evaluation focuses on the accuracy of a single best output. How is diversity quantified? Have the authors conducted a quantitative assessment of generative diversity—for example, for a given input image, how much variation is observed among the N geometry latents $\hat{Z}_{g}$ produced by the Prior (e.g., via Chamfer distance or LPIPS), and among different appearances $\hat{Z}_a$ generated by the Decoder conditioned on the same $\hat{Z}_g$?

---

> ### Author Response · Authors · 2025-11-20
> **Initial Response**
>
> Thank you for your nuanced understanding of our work, and for highlighting the effectiveness of CROCS and our hierarchical geometry-first approach. We are encouraged by the assessment that unPIC demonstrates "superior quality" and "robust generalization." We address the specific questions below.
>
> # 1. VAE KL Divergence (Question 1)
> We agree with the proposed hypothesis on why the CROCS VAE has significantly lower KL divergence than the RGB VAE (20823 vs 28005) before fine-tuning. CROCS pointmaps are fundamentally composed of smooth gradients (XYZ coordinates varying across the surface) and lack the high-frequency noise and complex texture patterns found in RGB images.
> This smoothness means the CROCS data distribution is naturally "closer" to the prior (standard Gaussian) and easier to compress, resulting in a lower KL penalty even before fine-tuning. The smoothness makes the CROCS latent space easier to navigate for a diffusion model and contributes to the high predictability we observe in the Geometry Prior.
>
> # 2. Inference Time Breakdown (Question 2)
> We provide the requested breakdown of the 1m 13s inference time on an A100 GPU (1000 steps):
> - Geometry Prior: ~36 seconds
> - Appearance Decoder: ~37 seconds
> - VAE Encoding/Decoding: <1 second (negligible)
> - Discussion: The computational cost is split almost evenly between the main modules. The Decoder is marginally slower due to the cross-attention mechanism processing the conditioning CROCS latents. As noted in the paper, we prioritized the conceptual framework over speed for this work, but the equal runtime split suggests that standard acceleration techniques (e.g., distillation, Flow Matching) applied to both modules would likely yield uniform speedups.
>
> # 3. From Point Cloud to Mesh (Question 3)
> We have explored converting our output point clouds into explicit meshes. Because unPIC produces dense point clouds with accurate geometry (derived from the smooth CROCS maps), standard meshing algorithms like Screened Poisson Reconstruction or Ball Pivoting work reasonably well. We will include meshed examples among the qualitative results we are putting together (as requested by reviewers Y436 & uYkA).
>
> # 4. Diversity (Question 4)
> We appreciate the suggestion to quantify diversity. Regarding the methodology, we would value the reviewer's input on the most appropriate metric.
> - **Latent space diversity:** As noted in Table 7, the Geometry (CROCS) VAE and Appearance (RGB) VAE arrive at different compression-vs-reconstruction trade-offs. The CROCS VAE has a significantly lower KL divergence and higher compression efficiency than the RGB VAE. Consequently, the raw variance of the geometry latents ($z_g$) is likely not directly comparable to that of the appearance latents ($z_a$).
> - **Output space diversity:** Calculating diversity in the decoded output space would be a bit more rigorous—for instance, computing the pairwise Chamfer Distance for geometry samples and pairwise LPIPS for appearance samples. Unfortunately, that would leave the geometric diversity incomparable with the appearance diversity.
> - **Question:** Given the representational differences in the VAEs, and the incomparability of output space metrics (Chamfer/LPIPS), would the reviewer agree that it might be preferable to look at the visual diversity of generated images?
>
> # 5. Camera Elevation (Weakness 1)
> We acknowledge the limitation regarding camera elevation. As noted in the review, our design choice prioritizes canonical target views (typical human mental rotation) but relies on implicit elevation inference. We agree that explicitly conditioning the model on an estimated or user-provided elevation angle (which could be stochastically dropped out during training to enable unconditional generation) would be a robust way to mitigate the "planar rotation" failure cases in future iterations.
>
> We hope these responses further clarify the technical details and merits of our work. Thank you once again.

---

> > ### Comment · Reviewer_BpYY · 2025-11-24
> >
> > Yes, if the author can measure the results using "the visual diversity of generated images", it would also be acceptable.

---

> > > ### Author Response · Authors · 2025-12-03
> > > **Thank you**
> > >
> > > Thank you for your continued positive assessment of our work, and for your suggestion to visualize the generative diversity of our method. We have uploaded a final revision, using the additional page allowed for the rebuttal to add a new figure to the main paper.
> > >
> > > **Figure 6** demonstrates the diversity of 3D outputs produced by unPIC. Crucially, it visualizes how our hierarchical approach captures multimodal distributions in both **geometry** (varying structural interpretations of ambiguous inputs) and **appearance** (texture variations). This provides qualitative evidence that unPIC is not merely regressing to a mean, but exploring the full range of plausible 3D shapes and textures consistent with the input image.

---

### Official Review · Reviewer_Y436 · 2025-11-01

**Soundness:** 3
**Presentation:** 3
**Contribution:** 3
**Rating:** 6
**Confidence:** 4

**Summary:**

The paper presents a framework for generating 3D-consistent novel views of an object from a single image. Unlike previous methods that jointly infer geometry and appearance, unPIC disentangles them in two stages: geometry prior and appearance decoding. In the first stage, the model predicts the object geometry using CROCS (Camera-Relative Object Coordinates), a dense, camera-aligned geometric representation. In the second stage, it decodes this geometry into multiview textured images, ensuring geometric consistency across different views. Moreover, the use of CROCS also enables direct reconstruction of 3D point clouds from the generated views.

**Strengths:**

- The paper introduces a simple yet sound idea that leads to clear performance improvements.
- The use of CROCS allows consistent 3D encoding without explicit segmentation or class priors, outperforming existing alternatives like NOCS.
- CROCS also allows direct extraction of 3D point clouds from generated views, simplifying 3D reconstruction pipelines by removing postprocessing.
- The experiments are extensive
  - Tab3 shows NVS results on 4 different datasets (Objaverse-XL, GSO, ABO, and DTC) and comparing with 6 baselines.
  - Tab5 and Fig5 demonstrate superior performance on 3D reconstruction.
  - The author also ablates the importance of geometry prior in Tab4.

**Weaknesses:**

- There are insufficient qualitative results showing the reconstructed 3D objects – either as point clouds or meshes. Only a single example is provided in Fig5.
- The quantitative results in Tab3, particularly the PSNR values, appear unusually high compared to the baselines. However, the qualitative examples (Fig4 and Supp.) do not seem to reflect such a large margin of improvement. This discrepancy raises concerns about how the metrics were computed and whether all methods were evaluated under the same viewing conditions.

**Questions:**

- Since some test images include camera elevation, is elevation provided or conditioned for the baseline methods?
- In Fig1, the authors refer to “input image(s),” suggesting that the method may accept multiple input views. However, the paper only presents results using a single input image. It would be helpful to clarify whether the proposed framework can be extended to multi-view inputs, and if so, how the model’s performance scales with additional views.

---

> ### Author Response · Authors · 2025-11-20
> **Initial Response**
>
> We thank the reviewer for their positive assessment and for recognizing the simplicity and effectiveness of the unPIC framework. We address the specific questions below.
>
> # 1. PSNR Discrepancy & Quantitative Results (Weakness 2)
> The gap in PSNR between unPIC and the baselines may indeed not be immediately obvious in the qualitative figures. This discrepancy stems from how pixel-wise metrics like PSNR penalize **pose misalignment.**
> - Many baselines (e.g., One-2-3-45, Free3D) generate plausible-looking textures but often suffer from geometric drift or slight pose inaccuracies (as shown in the "mutations" noted in Figure 4 and quantified by our IoU metric). In a pixel-wise comparison against a ground-truth target view, even a small geometric shift results in a significant PSNR penalty.
> - **unPIC’s advantage:** Because unPIC is **geometry-first**, it locks in the 3D structure (CROCS) before painting the texture. Our geometric supervision and lack of co-training help ensure that generated novel views are structurally aligned with the ground-truth target pose, leading to significantly higher PSNR scores.
> - **Evaluation protocol:** We **confirm that all methods were evaluated under the exact same conditions.** As stated in Sec 4.2, "For each test object, we sampled a random source and target view, and evaluated all models on the same pair".
>
> # 2. Qualitative 3D Results (Weakness 1)
> We will shortly provide qualitative examples of our 3D reconstruction results in comparison with InstantMesh and Direct3D in a common response to all reviewers.
>
> # 3. Camera Elevation (Question 1)
> We confirm that camera elevation was _not_ provided to the baseline methods (nor to unPIC).
> All methods were evaluated in a strictly "single image in" setting.
> Baselines that require camera parameters (like One-2-3-45) used their own internal estimation modules to infer elevation from the pixels.
> This ensures a fair comparison of each method's ability to infer 3D pose from 2D appearance implicitly.
>
> # 4. Multiview Inputs (Question 2)
> Regarding the "Input image(s)" label in Figure 1: unPIC is indeed designed to handle multiview inputs. As described in Sec 3.2, we treat inputs as a "masked set of K appearance latents". During training, we mask all but one arbitrary view, thus training unPIC as a single-view model.
> However, the masking mechanism indeed allows inference with K input views (placing them in the appropriate slots of the superimage) without architectural changes. We showed results with two input images in Figure 1, and a single input image in Figures 2, 4 & 5. Although our paper focuses on the single-view setting to establish the core capabilities of the CROCS representation, multiview fusion is a direct extension of this framework.
>
> We hope this clarifies the evaluation protocol and the architectural capabilities of unPIC. Many thanks for your time.

---

> > ### Author Response · Authors · 2025-12-03
> > **Thank you**
> >
> > Thank you for your early positive assessment of our work. While we regret that our conversation was cut short, we hope the additional 3D reconstruction results we shared would address your main concern (weakness 1), as they did for Reviewer uYkA. We trust that our initial response also helped answer your questions, and the second concern you raised about PSNR values (weakness 2).

---

### Official Review · Reviewer_uYkA · 2025-11-03

**Soundness:** 3
**Presentation:** 3
**Contribution:** 2
**Rating:** 4
**Confidence:** 5

**Summary:**

- The paper is clearly written, with many details deferred to a well-organized appendix.
- It fine-tunes a dedicated VAE for CROCS and reports a substantial VAE score improvement after fine-tuning.
- It presents extensive quantitative experiments and ablations, with results consistently favoring CROCS on the novel-view synthesis tasks.
- Code and checkpoints are open-sourced.

**Strengths:**

- The paper is clearly written, with many details deferred to a well-organized appendix.
- It fine-tunes a dedicated VAE for CROCS and reports a substantial VAE score improvement after fine-tuning.
- It presents extensive quantitative experiments and ablations, with results consistently favoring CROCS on the novel-view synthesis tasks.
- Code and checkpoints are open-sourced.

**Weaknesses:**

- Figure 5 is the only qualitative 3D reconstruction example; stronger qualitative evidence is needed. CROCS point maps may appear noisy on edges and thin structures, making denoising and detail preservation non-trivial; the resulting point cloud may be coarse, and vertex color aggregation across predicted views can be inconsistent, with non-trivial texture post-processing.
- Novel view synthesis is restricted to eight canonical views and cannot sample arbitrary viewpoints.
- Baselines are a little outdated; recent open-source SOTA (e.g., TRELLIS) is missing.
- The novelty of CROCS is limited: CROCS is adapted based on SpaRP’s NOCS variant (Xu et al., 2024a). Sec. 3.3 and Figure 3 explain the differences between the original NOCS and CROCS. However, both SpaRP’s NOCS variant and CROCS are oriented by the source camera’s azimuth and are axis-aligned; at the source view (Figure 3), CROCS is the same as SpaRP’s NOCS.

**Questions:**

- Please provide more qualitative examples and comparisons for 3D reconstruction results (point clouds) to demonstrate usefulness beyond canonical novel views.
- In the novel view synthesis experiments (Table 3), how are target views selected for each method? These baselines define different canonical target views than unPIC. Please clarify to ensure a fair comparison.

---

> ### Author Response · Authors · 2025-11-20
> **Clarification on unPIC vs. SpaRP**
>
> Many thanks for your detailed review and suggestions. We are preparing a comprehensive response including the requested qualitative comparisons (point clouds), which will follow shortly. In the meantime, we wish to provide a clarification on the distinction between SpaRP and unPIC. Our **novelty lies in the prediction target**:
>
> 1. **SpaRP** predicts NOCS for given views only (see their Fig 2):
>     - The goal is to establish correspondence between input images to estimate their camera poses (using PnP).
>     - Mechanism: It acts as _a discriminative tool_ to analyze existing visual data.
>
> 2. **unPIC** predicts CROCS for unknown views:
>     - The goal is to generate a "blueprint" for the 3D geometry of the entire object ($360^{\circ}$) from a single input view.
>     - Mechanism: It serves as _a generative prior_ to synthesize missing information.
>
> unPIC shows that it is not only possible to train a generative prior to predict coordinate maps for novel views (here we see a significant benefit from using CROCS over NOCS, see Table 1), but that it is **desirable to condition the generation of the final novel-view images on the expected geometry** (see our ablation in Sec 4.4). In contrast, SpaRP generates coordinate maps (NOCS) only for the given input views: in the single-image-to-3D setting, SpaRP would predict a single NOCS image to annotate the source view only.  SpaRP does not condition its novel-view generation on the inferred geometry (NOCS).
>
> We apologize this distinction was not clear in our Related Work (Sec 2, under Pointmaps). We will update it to make it clearer, contrasting unPIC’s generative use of coordinate maps for _unknown_ views against SpaRP’s discriminative use for _given_ views. Thank you very much.

---

> ### Author Response · Authors · 2025-11-27
> **3D Reconstruction, Target Views & Baselines**
>
> Thank you once again for your constructive review. Below, we address your remaining questions regarding 3D reconstruction quality, our evaluation protocol, and the baseline selection.
>
> # 1. 3D Reconstruction Results (Weakness 1 & Question 1)
> Please refer to our top-level comment titled "3D Reconstruction Results" and the updated Appendix C (we added Figure 11) in the _revised_ PDF. We provide 40 additional qualitative comparisons against Direct3D and InstantMesh. We visualize both our raw point clouds (directly converted from unPIC's CROCS predictions) and meshes (reconstructed via standard Screened Poisson).
> - These results demonstrate that unPIC’s point clouds effectively capture the full 3D structure—even for thin structures and complex geometries.
> - The point clouds are far from coarse (they contain up to 524k points each).
> - We do not see any obvious issues with vertex color aggregation, which would be easy to spot, e.g., as smudging of textures.
>
> We believe unPIC's geometric supervision and dedicated prior module are sufficient to ensure that the output point clouds are accurate and useful beyond canonical view synthesis.
>
> # 2. Target View Selection & Fairness (Weakness 2 & Question 2)
> We confirm that **all methods were evaluated on the exact same set of target views** to ensure a fair comparison.
>
> **Protocol:** Our test set consists of a 360-degree spin around the object's vertical axis at 45-degree increments ($K=8$ views, $θ_k \in \\{45, 90, 135, …, \\}$ degrees).
> - For unPIC: We use the model's native predictions at these fixed intervals.
> - For geometry-free baselines (e.g., One-2-3-45, Free3D): these models accept explicit camera pose inputs (R, T, or Plücker rays). We fed them the exact extrinsic matrices corresponding to the 45, 90, 135... degree rotations used by unPIC.
> - For mesh-based baselines (e.g., InstantMesh): we generated the 3D mesh and rendered it from the exact same target-camera poses using the default renderer in the open-source codebase.
>
> **Why these views?** As analyzed in **Appendix B.3**, we found that predicting these 8 canonical views is the core structural task for single-image-to-3D. The canonical views act as a "blueprint" that captures the object's geometric distribution. If a model can accurately predict these 8 sparse anchors, interpolating to arbitrary θ becomes significantly easier (this was also leveraged by CAT3D, whose sampling strategy was to predict 8 global views from a single image, then condition on 3 global views to generate “local” views). By fixing the target novel views for all models, we measure the model's ability to maintain global consistency (vital for 3D reconstruction) rather than just novel-view plausibility.
>
> # 3. Baseline Selection (Weakness 3)
> We note that unPIC has already been benchmarked against a comprehensive suite of 7 leading methods (CAT3D, EscherNet, Free3D, OpenLRM, InstantMesh, One-2-3-45, and Direct3D).
>
> While we acknowledge TRELLIS (CVPR 2025) as impactful recent work, the **scientific contribution of our paper goes beyond leaderboard performance.** We demonstrate that a geometry-driven hierarchical pipeline is superior to a geometry-free pipeline for multiview diffusion. Removing the geometry prior (while keeping the rest of the unPIC pipeline identical) leads to a substantial drop in performance in our ablation study in **Table 4**.
>
> We hope this clarifies our experimental setup and demonstrates the robustness of our results. Thank you again for your valuable time.

---

> > ### Comment · Reviewer_uYkA · 2025-11-28
> >
> > **3D reconstruction quality**
> >
> > The new examples in Appendix C (incl. Fig. 11) are actually much clearer than what was originally shown. The denser point clouds and side-by-side comparisons help a lot, and I think including some of these examples in the main paper would make the work look stronger. While the method is still not comparable to current SOTA methodds, the updated results are reasonable for this approach. With this in mind, I’m willing to raise my score to 6.
> >
> > **Evaluation protocol / fairness**
> >
> > That said, I still respectfully disagree with your justification regarding the novel-view evaluation. In your setup, you compare your canonical blueprint views—which your model is explicitly optimized to produce—against other methods’ mesh renderings or arbitrary novel views under the same camera poses. This is not symmetric. Image-based predictions and mesh renderings behave very differently, and mixing them in this way inevitably favors your method.
> >
> > A simple example: if InstantMesh were allowed to define its own canonical “blueprint” views from its first-stage outputs, and we compared those directly to your mesh/point-cloud renderings, the outcome would likely look very different. So the issue is not the specific camera angles, but the fact that you are comparing your native canonical predictions to another method’s rendered geometry and non-canonical predictions. This is, in my view, still not a fair comparison.

---

> > > ### Author Response · Authors · 2025-12-03
> > > **Thank you**
> > >
> > > Thank you for carefully going through the additional 3D reconstruction results we shared. We are glad to hear that the additional results have addressed your main concern; we appreciate your decision to raise your assessment to a 6.

---

### Author Response · Authors · 2025-11-25
**3D Reconstruction Results**

As requested by Reviewers uYkA and Y436, we have uploaded a revised PDF (see **Appendix C / Figure 11**) and a supplementary zip file containing extensive qualitative 3D reconstruction results. We provide **40 additional examples** comparing unPIC’s point clouds and meshes against Direct3D and InstantMesh.

# 1. Direct Point Cloud Extraction (Raw Output)
unPIC produces dense point clouds ($K \times W \times H = 8 \times 256 \times 256 \approx 524k$ points) by directly converting the generated CROCS and appearance images. As shown in Table 5, these raw point clouds achieve better geometric accuracy (Chamfer distance) than two SoTA methods (Direct3D and InstantMesh).

Note on visualization: The raw point clouds may exhibit some high-frequency outliers due to the background removal process. However, as our quantitative metrics and mesh reconstructions show, the underlying geometric structure remains highly accurate.

# 2. Mesh Reconstruction (No Neural Networks)
To demonstrate the structural integrity of our predictions, we converted our raw point clouds into meshes using standard geometric processing without any learnable parameters or neural refinement:
- We used pymeshlab's `compute_normal_for_point_clouds(smoothiter=3)` for normal estimation.
- We then ran `generate_surface_reconstruction_screened_poisson(preclean=True)` [1] to extract the final mesh.

The success of these standard algorithms verifies that unPIC's generated geometry is coherent and robust, addressing Reviewer uYkA's query regarding edge noise and detail preservation.

# 3. Supplementary Details
- **PDF (Appendix C):** Visual comparisons of 40 objects across unPIC (point clouds + meshes), Direct3D, and InstantMesh.
- **ZIP File:** Contains the raw `.ply` point cloud files for all 40 examples. Due to the 100MB supplementary limit, we could only include high-resolution `.obj` meshes for a subset of 4 examples. The mesh files are significantly larger than the point clouds due to the additional storage required for the connectivity (faces) and high vertex density preserved by the Poisson reconstruction.
- Viewing: We recommend viewing meshes in MeshLab or similar dedicated software to ensure correct lighting and orientation.

[1] Kazhdan, Michael, and Hugues Hoppe. "Screened poisson surface reconstruction." ACM Transactions on Graphics (ToG) 32.3 (2013): 1-13.

Many thanks for your attention to our work.

---

### Author Response · Authors · 2025-12-01
**Summary of Rebuttal & Note on Reverted Scores**

Dear Area Chair,

Thank you for taking on our submission under extraordinary circumstances as part of your new batch of papers. Given that all reviewer scores have been reverted to their pre-rebuttal state, and that reviewer interaction has been disabled, we are writing to summarize the consensus reached during the discussion period.

Specifically, we wish to highlight that **two reviewers explicitly stated their intent to raise their scores** based on our new data and clarifications. We hope you will consider these "lost updates" in your final assessment:

# 1. Reviewer uYkA (Original Score: 4 → Stated Intent: 6)

- **Concern:** The reviewer initially felt the qualitative 3D reconstruction results were insufficient.
- **Resolution:** We uploaded a revised pdf (Appendix C) and supplementary zip file containing 40 additional comparisons against Direct3D and InstantMesh.
- **Outcome:** On Nov 28, Reviewer uYkA commented: _"The new examples... are actually much clearer... With this in mind, I’m willing to raise my score to 6."_

# 2. Reviewer Um9x (Original Score: 2 → Updated: 4)
- **Concern:** The reviewer initially believed the method lacked novelty compared to SweetDreamer and MVDream.
- **Resolution:** We clarified that unPIC is a feed-forward, geometry-first generative model (using camera-relative CROCS), whereas SweetDreamer relies on optimization and object-centric coordinates.
- **Outcome:** On Nov 20, Reviewer Um9x commented: _"I understand the difference... I could improve my score to borderline reject."_ They acted on it by increasing their official score on Nov 21, which was only three days into the rebuttal period, signaling that they were perhaps open to being convinced further.

# 3. Reviewer BpYY (Score: 8)
- Remains positive and accepted our proposal for showing diversity visually. We are preparing a revised pdf with a new figure in time for the deadline on Dec 3.

## Summary of Changes
To support the discussions above, the current version of the paper includes:
- **Appendix C / Figure 11:** Extensive qualitative comparisons of point clouds and meshes (addressing uYkA and Y436).
- **Table 1 & Section 4.1:** Clarifications on the performance gap between our camera-relative approach and object-centric baselines (addressing Um9x).

We are disappointed that the official system cannot reflect the consensus we worked hard to reach with the reviewers, but we trust that this summary helps you navigate the discussion thread efficiently.

Sincerely,

The Authors

---

> ### Author Response · Authors · 2025-12-03
> **Summary of Final Updates**
>
> We have used the optional 10th page to:
> - Add a new figure (**Figure 6**), visualizing the diversity of unPIC’s outputs as requested by Reviewer BpYY.
> - Expand our **Related Work**, clarifying unPIC’s novelty in relation to prior work raised by Reviewers Um9x and uYkA.
> - Move examples showing unPIC’s generalization to in-the-wild images from Appendix C to **Figure 7** in the main paper. We wanted to emphasize this point, as we believe the generalization was only highlighted by Reviewer BpYY (rating: 8)--we searched for terms like ‘generalization’, ‘wild’, and ‘real’ across all reviews.
>
> Many thanks for your attention to our work.

---

### Meta-Review · Area_Chair_wJU2 · 2026-01-03

**Summary:**

The reviews present a mixed evaluation, with high-confidence reviewers (confidence ≥ 4) generally leaning toward rejecting the paper. The AC conducted a thorough review of both the discussions and the paper, identifying several concerns, including issues with evaluation protocols related to PSNR and the lack of comparisons to critical baselines like TRELLIS. It also noted insufficient justification for the novelty of the CROCS component. Additionally, the authors used evasive language in their rebuttal that did not adequately address these key issues.

The AC recommends rejecting the paper and encourages the authors to incorporate feedback, conduct fair comparisons with leading baselines, comprehensively cite and discuss relevant works, and moderate claims regarding the technical novelty of CROCS.

**Reviewer Concerns:**

- Reviewer uYkA: The response to the urge for "more qualitative results" is solid. However, sampling arbitrary views requires an additional model like CAT3D, which cannot genuinely support this claim. The rebuttal still lacks a comparison with TRELLIS, a key work in image-to-3D, and the explanation of "going beyond leaderboard performance" feels weak and somewhat evasive. Moreover, CROCS closely resembles SpaRP’s NOCS variant; the difference—"given vs. unknown views"—is merely an architectural choice, not a distinction in shape representation.

- Reviewer Y436: The issue of "insufficient qualitative results" is well addressed. However, the explanation for "PSNR appears unusually high" lacks convincing evidence. PSNR behaves differently for mesh rendering and generated views; mixing these in evaluations unfairly favors this work.

- Reviewer BpYY: Concerns are well addressed.

- Reviewer Um9x: After reviewing the discussion, AC finds that the concern about "limited novelty" remains inadequately addressed. While the authors detail technical differences among the missing competitors, they fail to convincingly justify the novelty of CROCS. Since CROCS is the main contribution, this evasive response underscores its technical limitations.

**Reviewer Scores:**

- Reviewer uYkA (conf 5): is likely to maintain or even lower the score (2 → 4). Note that uYkA stated "raise my score to 6" on November 28, yet the issue of "evaluation protocol/fairness" is actually a serious complaint, echoing Y436's concern about "PSNR appearing unusually high."

- Reviewer Y436 (conf 4): is likely to maintain or even lower the score (4 → 6).

- Reviewer BpYY (conf 3): is likely to keep the score (8).

- Reviewer Um9x (conf 5): is likely to maintain or raise the score (2 → 4).

---

> ### Public Comment · ~Rishabh_Kabra1 · 2026-03-13
> **Factual error in meta-review**
>
> The meta-review incorrectly notes that our initial scores were 2/2/4/8 (average 4.0) when in fact they were 2/4/6/8 (average 5.0). This is easily verified below, as all ICLR reviews have been reset to the pre-rebuttal scores.
>
> During the rebuttal, the reviewers committed to raise their scores to 4/6/6/8 (average 6.0). This was also incorrectly reported in the meta-review (predictions 4/4/6/8, averaging 5.5).
>
> Unfortunately the final decision goes contrary to the consensus we worked to reach with the reviewers.

---

### Decision · Program_Chairs · 2026-01-26

Reject